# A high-entropy manganite in an ordered nanocomposite for long-term application in solid oxide cells

F. Baiutti [1✉], F. Chiabrera [1], M. Acosta [2], D. Diercks [3], D. Parfitt[4], J. Santiso [5], X. Wang[6], A. Cavallaro [7], A. Morata[1], H. Wang [6], A. Chroneos[4], J. MacManus-Driscoll [2] & A. Tarancon [1,8✉]

The implementation of nano-engineered composite oxides opens up the way towards the development of a novel class of functional materials with enhanced electrochemical properties. Here we report on the realization of vertically aligned nanocomposites of lanthanum strontium manganite and doped ceria with straight applicability as functional layers in high-temperature energy conversion devices. By a detailed analysis using complementary state-of-the-art techniques, which include atom-probe tomography combined with oxygen isotopic exchange, we assess the local structural and electrochemical functionalities and we allow direct observation of local fast oxygen diffusion pathways. The resulting ordered mesostructure, which is characterized by a coherent, dense array of vertical interfaces, shows high electrochemically activity and suppressed dopant segregation. The latter is ascribed to spontaneous cationic intermixing enabling lattice stabilization, according to density functional theory calculations. This work highlights the relevance of local disorder and long-range arrangements for functional oxides nano-engineering and introduces an advanced method for the local analysis of mass transport phenomena.

[1] Catalonia Institute for Energy Research (IREC), Jardins de Les Dones de Negre 1, Sant Adrià del Besòs, Barcelona, Spain. [2] Department of Materials Science and Metallurgy, University of Cambridge, Cambridge, UK. [3] Department of Metallurgical and Materials Engineering, Colorado School of Mines, Golden, CO, USA. [4] Faculty of Engineering, Environment and Computing, Coventry University, Coventry, UK. [5] Catalan Institute of Nanoscience and Nanotechnology, ICN2, CSIC and The Barcelona Institute of Science and Technology (BIST), Campus UAB, Bellaterra, Barcelona, Spain. [6] School of Materials Engineering, Purdue University, West Lafayette, IN, USA. [7] Department of Materials, Imperial College London, London, UK. [8] ICREA, 23 Passeig Lluís Companys, Barcelona, Spain. ✉email: fbaiutti@irec.cat; atarancon@irec.cat

Solid Oxide Cells (SOCs), which are capable of reversibly converting chemical to electrical energy with high theoretical efficiency and a minimized footprint[1,2], can play a pivotal role in the widespread implementation of next-generation energy devices. By means of modern fabrication technologies, SOCs can take advantage of beyond-state-of-the-art concepts for materials' nanoengineering, this way achieving enhanced performance and further downscaling[3]. To date, however, sluggish electrode catalytic activity, poor thermochemical stability, and limitations in the effective implementation of advanced nanoscale materials, have hindered the scalability of SOCs technology and its viability as a portable energy source[4]. In order to improve the performance of SOC air electrode without using expensive and rare noble metals, the development of functional ceramic nanostructures based on mixed ionic-electronic conducting oxides (MIEC) represents a promising approach[5]. In MIEC materials (e.g., doped perovskite manganites, cobaltites, ferrites), a good trade-off between the free energy of the oxygen redox cycle and the enthalpy of oxide reduction can be achieved, ensuring high electrocatalytic activity on the whole free surface together with fast oxygen ion transport[6]. Most importantly, when MIECs are employed in the form of thin films, the dense microstructure allows for excellent in-plane percolation, while the possibility of nanostructuring via cutting-edge thin film techniques opens up the path toward the exploitation of nanoscale effects such as local fast oxygen reduction kinetics and transport[7]. Although the implementation of MIEC-based thin films for the fabrication of SOC functional layers is of great potential e.g., for avoiding the occurrence of areas with high electronic current density or high oxygen chemical potential gradients which may potentially lead to device failure[8], intrinsic limitations still occur especially in relation to a poor thermal stability owing to dopant segregation toward the surface. Limiting cationic migration is of paramount importance in order to avoid the possible formation of insulating secondary phases and to maintain the surface activity over time[9,10]. State-of-the-art strategies for enhancing MIEC thin film performance comprise strain engineering via epitaxial growth[11,12], chemical doping[13], introduction of an interface (so-called *heterogeneous doping*)[14–16], or surface decoration[17]. Such methods, however, are often limited to model systems with no technological applicability, and very few of them have proven to be successful in simultaneously improving the initial oxygen reduction reaction (ORR) kinetics while mitigating detrimental surface cationic segregation[18].

In order to enhance the mass transport kinetics at intermediate temperatures, MIEC are combined to pure oxygen conductors to form (nano)composites, which find wide application not just as SOCs electrodes, but also as dual phase membranes for oxygen separation owing to the ability of transporting both ionic and electronic species while maintaining structural stability[19–21]. Classical approaches for composite fabrication lead however to the formation of disordered structures, which suffer from thermal instabilities and limited percolation for mass and charge transport[22,23]. A highly potential alternative for the fabrication of nanocomposite electrode functional layers is represented by vertically aligned nanostructures (VANs), which are characterized by an off-plane, nanoscale phase alternation[24]. Nano-engineered VANs have been extensively investigated in the last years: Model systems, generally characterized by 'nanocolumn in matrix' mesostructures (domain-matching epitaxy)[25,26], have revealed unexpected properties including e.g., enhanced ionic conductivity[27], controlled resistive switching[28] and improved photocatalytic activity[29]. Such functionalities have been ascribed to the entanglement of interface effects including elastic strain and ionic or electronic redistribution[30]. Some reports also deal with effective nanocomposite layers or interlayers for

electrochemical applications: Yoon et al[31]. and Cho et al[32]. proposed similar solutions, based on self-assembled MIEC-fluorite thin film vertical nanocomposites, for application as functional layers in fuel cells, while Ma et al[33]. studied the fundamental properties of Co-based VANs, highlighting enhanced ORR kinetics. Although such works demonstrate the great potential of the VAN approach, to the best of the authors' knowledge, so far inert model substrates had to be employed for the realization of ideal structures, while the use of "non-ideal" supports such as real electrolyte layers irremediably led to the formation of arrangements with higher disorder (i.e., incoherent interfaces and low interface density), possibly inhibiting the full exploitations of VANs in real electrochemical devices. Notably, the possibility of fabricating superior MIEC nanostructures by design goes hand-in-hand with the development of techniques which are capable of capturing nanoscale phenomena with higher accuracy.

In the present work, we describe the fabrication of a self-assembled thin film VAN nanocomposite as an ad hoc-engineered MIEC. The nominal constituting phases are a mainly electronic conductor such as $La_{0.8}Sr_{0.2}MnO_3$ (LSM) and an ionic conductor, namely $Ce_{0.8}Sm_{0.2}O_2$ (SDC). The structure possesses long-range order, out-of-plane epitaxy and intimate phase alternation (column width $\approx 10$ nm) for maximized triple-phase boundary density ($>10^6$ cm cm$^{-2}$). The fabrication uses an yttria-stabilized zirconia (YSZ) electrolyte as a support—we stress here that such a support is considered as an ideal platform for the widespread implementation of functional oxide nanomaterials in devices[34]—and results in the combination of key features for high-temperature electrochemical applications: accelerated ORR kinetics and, most importantly, high stability against temperature-driven cationic migration. The nanostructure has therefore straight applicability as a SOC functional electrode layer. The system is also ideal for implementing complementary state-of-the art and novel techniques for locally assessing structural and functional aspects. Direct observation of a local fast mass transport pathway along the SDC matrix (<10 nm wide) has been obtained by isotopic oxygen labelling combined with atom-probe tomography (APT), achieving superior resolution for describing oxygen diffusion.

## Results and discussion
**Structural characterization highlights long-range order.** A sketch of the LSM-SDC VAN mesostructure, composed by LSM pillars in an SDC continuous matrix, is depicted in Fig. 1 (panel a), together scanning transmission electron microscopy (STEM—panels b–e) and high-resolution x-ray diffraction (XRD—panel f). In the lower-magnification high-angle angular dark field (HAADF) STEM image of a thin cross-sectional lamella (Fig. 1b), one can appreciate the thin columnar, long-range ordered, mesostructure for large portions of the film (column width is as narrow as $\approx 5$ nm, length $\approx 100$ nm, no tortuosity). One can also see that no porosity is present. The precise sequence of alternating (La, Sr, Mn)- and Ce-based phases can be clearly identified in Fig. 1c, where a HAADF image and an Energy-dispersive X-ray spectroscopy (EDS) color maps of the area highlighted in Fig. 1b are shown. In the HAADF image, owing to the higher $Z$ numbers of Ce and Sm, SDC columns have higher contrast than those of LSM. Interestingly, the color map underlines not only the clear separation between the LSM columns (dominated by Sr, Mn) and the SDC matrix (Ce, Sm), but also a certain degree of cationic intermixing especially between La, Sm and Ce. A quantitative and theoretical analysis on cationic redistribution will be discussed in detail in later paragraphs. In Fig. 1c, it is also interesting to notice that the substrate-film interface is characterized by an even more pronounced intermixing and that no phase separation is

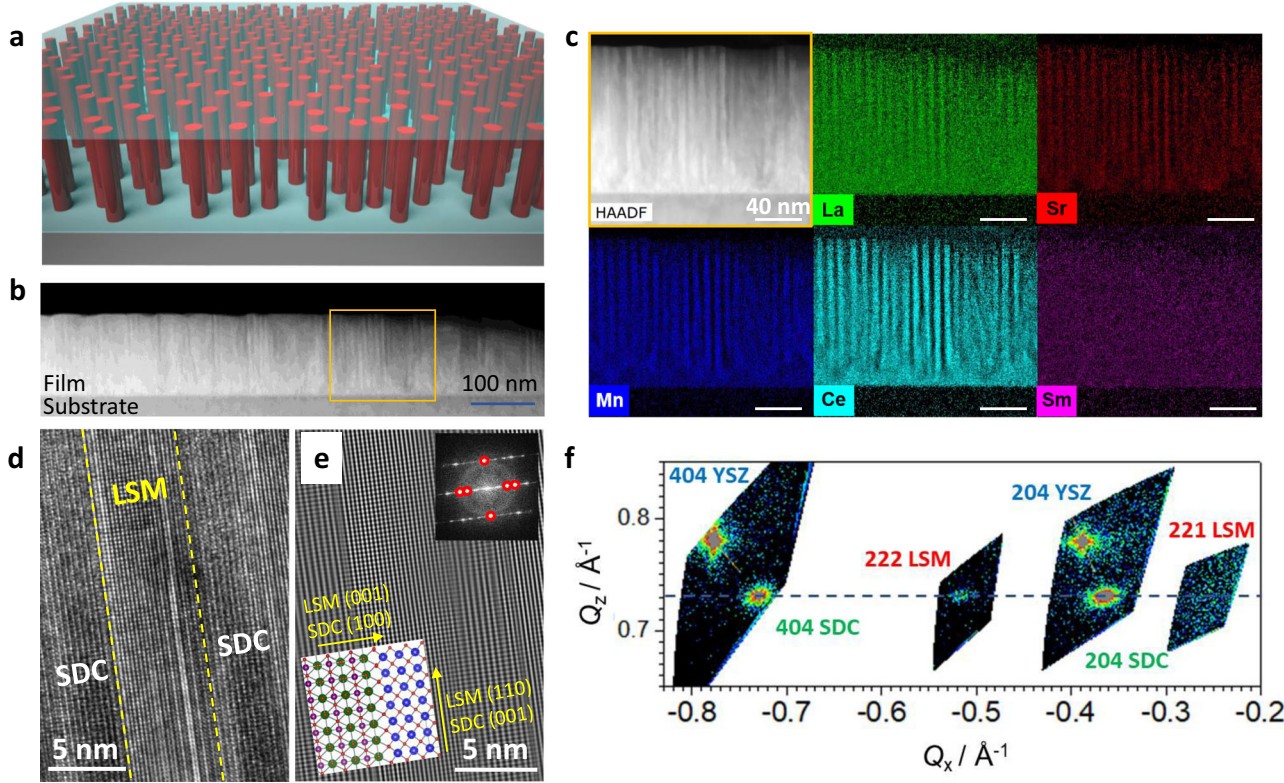

**Fig. 1 Structural investigation of LSM-SDC. a** Sketch of the LSM-SDC VAN structure. **b** Low-magnification image of the film, highlighting the long-range order of the pillar structure. The thickness decrease on the right side should be ascribed to TEM lamella preparation. **c** HAADF image and EDS colour maps for a representative columnar region of the film. **d** HR-TEM of the alternating SDC and LSM columnar structure. **e** IFFT of (**d**). The atomic ordering and the epitaxial relationship between the LSM (out-of-plane [110], zone axis [1-10]) and the SDC (out-of-plane [001], zone axis [100]) regions is clearly visible, as schematically sketched by the bottom inset. In the top inset, the original FFT and the filtered reflections (highlighted by the red circles) are shown. **f** RSM of the out-of-plane matching (001)-oriented SDC domains (404 and 204 SDC reflections) and (110)-orientedLSM domains (222 and 221 reflections). The 404 and 202 YSZ substrate reflections are also visible. The dotted horizontal line passes through the maxima of the LSM and SDC reflections, highlighting perfect out-of-plane lattice match between the two phases. Please note that the structural investigation was carried out after thermal ageing (700 °C, 100 h).

apparent. We ascribed such a finding to the structural (lattice match) and chemical (wettability) properties of the fluorite YSZ substrate which favors the formation of a single, Ce-rich, phase during the first stages of film growth.

The HR-TEM of a representative columnar region is shown in Fig. 1d, where one can notice a perfect local order. The atomic arrangement can be further analyzed in Fig. 1e (inverse filtered fast Fourier transform (IFFT) of the region in Fig. 1d): A structurally coherent relationship is established out-of-plane ($z$-direction) between the two phases. Also, no evidence of other phases or of spurious orientations are detected in these regions. The in-plane and out-of-plane orientation relationships between the film phases and the substrate in such highly ordered majority portions are LSM (001) || SDC (100) || YSZ (100) and LSM (110) || SDC (001) || YSZ (001), respectively, as sketched in the bottom-left inset of panel e. A dedicated XRD analysis by means of reciprocal space mapping (XRD-RSM) of the matching planes in the $z$-direction i.e., (110) for LSM and (001) for SDC is shown in Fig. 1f. Here, the alignment of the LSM and SDC reflection in $z$ confirms the lattice match epitaxy in the out-of-plane direction at the global scale. Interestingly, the resulting lattice spacing for LSM ($d_x = 3.874$ Å and $d_z = 2.739$ Å for the in-plane and for the $z$-direction, respectively, vs $d = 3.906$ Å for bulk LSM in pseudocubic notation) indicates an isotropic compressive deformation of the cell ($-0.82\%$). (Please note here that $d_z$ refers to the (110) direction and the corresponding out-of-plane lattice parameter is $d_z \cdot \sqrt{2} = 3.874$ Å.) The reduced lattice parameter is consistent with cationic substitution (Ce and Sm replacing La—cf.

Fig. 1c), in agreement with previous reports and as also confirmed by the simulations presented later in the text (see below DFT section)[24]. Please refer to Supplementary Note 1 for the complete XRD characterization of the films.

**Observing oxygen diffusion pathways and cationic intermixing by APT.** Complementary to HR-TEM, the use of Atom Probe Tomography (APT) allowed us to obtain meaningful insights on the structure chemistry and on the oxygen kinetics, as summarized in Fig. 2. The technique, which is based on the Time-of-Flight (ToF) analysis of atoms evaporated through a pulsed laser impinging a highly biased sample tip, possesses unique capabilities for 3D imaging and precise quantitative chemical composition analysis with sub-nm resolution[35]. In Fig. 2a, b, projections of the spatial distribution of Mn (red—representative of LSM) and Ce (light blue—for SDC) over a sample tip (HAADF-STEM in Supplementary Fig. 2) are reported. Consistently with the TEM analysis (cf. Fig. 1), one can observe the presence of well-defined LSM pillars, immersed in an SDC matrix. The composition of the two phases has been carefully analyzed along a region crossing multiple LSM and SDC domains (blue cylinder in Fig. 2a, b). The resulting elemental distribution is reported in Fig. 2c and confirms the modulation of Mn, Sr and Ce when crossing each phase. One challenge in APT analysis of multiphase materials is that differences in the evaporation fields of the materials can result in changes in the local radius of curvature near the interfaces of the phases, which produce ion

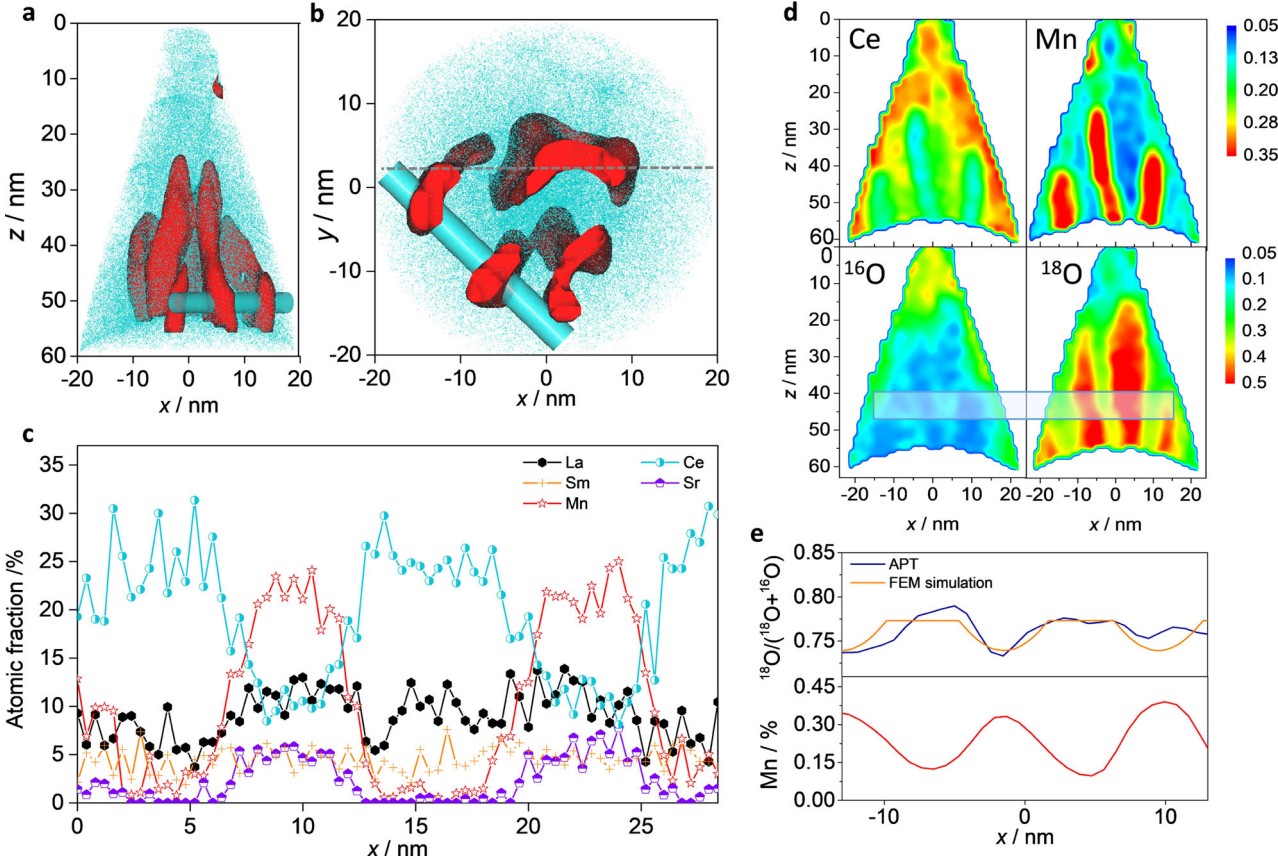

**Fig. 2 APT on LSM-SDC.** Lateral (**a**) and bottom view (**b**) 3D renderings, highlighting the areas of LSM (red surface) and SDC (light blue dots) using a 30 at% La+Sr+Mn isoconcentration surface. **c** Cationic atomic fraction along the linescan indicated by the blue cylinder in (**a**) and (**b**). **d** Concentration color maps for Ce, Mn, $^{18}O$ and $^{16}O$ from the slice indicated by the gray dotted line in (**b**). **e** Top: Experimental (blue—obtained upon integration of the area in the rectangle in (**d**) and simulated (red) $^{18}O$ fraction profiles. Bottom: Mn concentration profile.

**Table 1 Elemental atomic fractions in LSM and SDC.**

|            |     | **LSM** |             |     | **SDC** |             |
|------------|-----|--------|-------------|-----|---------|-------------|
|            |     | Counts | Cat.%       |     | Counts  | Cat.%       |
| A-site LSM | Mn  | 22790  | 43.5 ± 0.5  | La  | 50705   | 20.9 ± 0.7  |
|            | Ce  | 8227   | 15.7 ± 0.6  | Mn  | 12420   | 5.1 ± 0.7   |
|            | La  | 11374  | 21.7 ± 0.6  | Sr  | 2610    | 1.1 ± 0.2   |
|            | Sr  | 6588   | 12.6 ± 0.5  | Ce  | 150310  | 61.9 ± 0.8  |
|            | Sm  | 3389   | 6.5 ± 0.4   | Sm  | 26978   | 11.1 ± 0.4  |

These were calculated from the core regions of each phase, excluding the interfacial regions. The uncertainty values are taken from dividing the LSM phase into 7 regions and the SDC phase into 30 regions and taking the standard deviations of the compositions from each of the individual regions.

trajectories that deviate from that presumed in generating the reconstruction of the analyzed volume[36]. The result is apparent intermixing of the higher evaporation field phase (SDC in this case) into the lower evaporation field phase (LSM). Previous research has indicated that the extent of this overlap is generally up to about 2 nm around the phase interfaces and that for regions larger than 2 nm, the "core" of the lower evaporation field phase that excludes the outer 2 nm of material retains the accurate composition of that phase[36,37]. Even with this complication, the non-negligible intermixing of the other cations is clear (in agreement with the TEM observations—cf. Fig. 1) and results in the substitution of La by Ce and Sm in LSM (15.7 ± 0.6 and 6.5 ± 0.4 cat. %, respectively), while a large La content is found in SDC (20.9 ± 0.7 cat. %). The measured elemental atomic fraction is reported in Table 1. Please refer to Supplementary Note 2 for

more details on the method. In Supplementary Table 1, we also report the results on the quantification of atomic fractions for single phase LSM and SDC films that we performed in order to assess the accuracy of the APT method under different measurement (laser energy) conditions. We note here that, while TEM imaging and spectroscopy result in a direct phase/composition analysis on large portions of a cross-sectional specimen, APT is able to provide complementary local information via a 3-dimensional reconstruction, with enhanced ability to quantitatively assess local elemental distributions.

While the observed level of intermixing is expected to have only a limited impact on the oxygen mobility in SDC owing to dopant excess (≈32 at.% Sm+La)[38], the replacement of La in LSM by Ce and Sm plays a key role, according to our theoretical calculations, in the outstanding thermal stability of the structure as

discussed later (see below DFT). As reported previously and as resulting from steric considerations, Ce is expected to be mostly present as an isovalent substitutional dopant for La in LSM, thus not direcly affecting the electrochemical transport properties[39].

Taking advantage of the highly ordered and dense microstructure of our VANs, we employed APT as a techinque for analyzing the electrochemical functionalites of the VAN functional layer at the nanoscale (oxygen mass transport properties) via direct observation of oxygen diffusion pathways in VANs exchanged with oxygen isotopes ($^{18}O$) under controlled conditions of temperature and time. The sensitivity of APT toward oxygen isotopes has been demonstrated previously[40]. Unlike conventional Isotope Exchange Depth Profiling (IEDP) by ToF Secondary Ion Mass Spectrometry[41], which is commonly employed in bulk and homogeneous materials, our approach is characterized by a greatly enhanced lateral resolution down to the <10 nm-level (vs > 100 nm for IEDP)[42]. Figure 2d represents a snapshot of the oxygen incorporation in a LSM-SDC VAN after 6000 s of exchange with labelled oxygen at $T = 550\,°C$, showing concentration maps of $^{18}O$ and $^{16}O$ oxygen atoms, alongside the different cations, as obtained from APT. In the top panels, one can immediately recognize the LSM pillars, indicated by the presence of high Mn concentration areas, surrounded by a ceria matrix. In the bottom panels, areas of maximum $^{18}O$ concentration, corresponding to oxygen incorporated during the exchange, are clearly correlated to the SDC matrix while LSM pillars present much lower concentration. This provides direct display of the presence of a preferential local oxygen diffusion pathway (along the nm-wide ceria regions), after the incorporation through the active surface (top of the APT tip in Fig. 2), proving the efficacy of the proposed VANs in combining the excellent ionic conductivity of SDC with the good oxygen affinity of the LSM (typically hindered by its poor oxygen diffusivity). The effect of materials combination is here maximized by the unique VAN microstructure, which is characterized by no tortuosity for off-plane mass transport and by high triple-phase boundary density (>$10^6$ cm·cm$^{-2}$, cf. Fig. 1) for oxygen incorporation. Notably, the reported direct observation of fast local oxygen pathways, which is demonstrated here in the case of VANs, has wide applicability and represents a decisive step forward to the local analysis of oxygen incorporation in nanomaterials, for which singularities are often found at the nm-level (e.g., fast grain boundary oxygen diffusion)[43]. In order to quantitatively assess this picture, the measured diffusion profiles (integrated over the area between the dotted lines in Fig. 2d) were fitted using Finite Element Method (FEM) simulations on a modelled 3D geometry consisting of LSM columns (with slow oxygen diffusivity) embedded in an SDC matrix (further details on the modelling are given in Supplementary Note 3). Please note that the experimental data do not suggest any singularity in the oxygen diffusivity at the interface, either because not present or because of a very limited effect on the behavior of the VAN thin film. For these reasons, the FEM model does not include any nanoscopic parameters for describing oxygen transport at the LSM/SDC interface, which is modeled as an abrupt change of diffusivity in the two materials (continuous layer)[44]. The comparison between the measured and the fitted ($^{18}O/(^{18}O + ^{16}O)$) profiles are represented in Fig. 2e, top panel. The fitting of the experimental profiles allowed extraction of the oxygen surface exchange coefficient of the composite ($k^* = 2.4 \cdot 10^{-9}$ cm s$^{-1}$) and the diffusion coefficient of oxygen in LSM ($D^* = 3.6 \cdot 10^{-17}$ cm$^2$·s$^{-1}$) at 550 °C (note that no information on the ion diffusivity is accessible by tracer method for SDC since these regions saturate owing to the fast transport kinetics). By comparing the tracer fraction and the Mn concentration profiles (Fig. 2e, bottom panel), one can also observe that an $^{18}O$ concentration decay is visible in LSM (maximum concentration

at the LSM-SDC interface), as a direct consequence of lateral oxygen diffusion from the SDC, while vertical transport in LSM is negligible (cf. also Supplementary Note 3). Similar results on a different 2D cross-section are reported in Supplementary Fig. 4. The resulting surface exchange coefficient of the VAN is remarkably higher than the one of LSM ($k^* \sim 1 \cdot 10^{-11}$ cm$^2$·s$^{-1}$) and also than the one reported for related LSM-YSZ dense composites ($k^* \sim 2 \cdot 10^{-10}$ cm$^2$·s$^{-1}$) at the same temperature[45,46]. As far as the diffusion coefficient of LSM is concerned, this is fully consistent electrochemical measurements by impedance spectroscopy ($D^*_{impedance} = 5.4 \cdot 10^{-17}$ cm$^2$·s$^{-1}$) and with literature data (cf. following section and Supplementary Fig. 8).

**Electrochemical performance.** Complementary electrochemical characterization was carried out by impedance spectroscopy (EIS) in conventional two-electrode supported electrolyte cells (Au-LSM-SDC VAN/YSZ/Ag) and is summarized in Fig. 3.

In Fig. 3a, exemplary Nyquist plots for LSM-SDC and analogous LSM cells are reported ($T = 750\,°C$, air atmosphere). LSM exhibits the typical polarization arc of MIEC materials stemming from a co-limited process, i.e., ambipolar diffusion ($\approx 45°$ straight element at high frequencies) and ORR (semicircle at low frequencies)[47]. One can appreciate that in LSM-SDC both contributions are drastically reduced (cf. also Supplementary Fig. 9 for comparison of the area-specific resistance—ASR). The polarization arc exhibits an almost perfect impedance arc— indicating that the oxygen incorporation is limited by the surface reduction reaction only—while the mass transport contribution is completely suppressed. This is fully consistent with the APT characterization of the oxygen kinetics highlighting fast vertical diffusion of oxygen through the SDC phase and is ascribed to the intrinsic fast ionic conductivity of the fluorite, together with the advantageous off-plane VAN geometry. Nyquist plot for single phase SDC cells, exhibiting much higher ASR, is reported for comparison in Supplementary Fig. 10.

By adjusting the EIS spectra of the cells with a physically meaningful equivalent circuit (i.e., a modified Jamnik-Maier circuit for LSM and a simple ZARC element for LSM-SDC— please refer to Supplementary Note 4 for further details on impedance spectroscopy analysis), the effective rate constant for the ORR $k^q$ can be obtained: $k^q = k_b \cdot T / R_s \cdot c_{O_2} \cdot z_i^2 \cdot e^2$, being $R_s$ the surface resistance for the oxygen chemical reaction[48]. The results are reported in Fig. 3b, where LSM-SDC is put in direct comparison with (i) stoichiometric LSM, (ii) LSM-YSZ dense composites and (iii) state-of-the-art cobalt-based MIEC for intermediate-temperature electrode applications (La$_{0.8}$Sr$_{0.2}$CoO$_3$—LSCo82)[45,46]. One can see that the oxygen incorporation kinetics are remarkably enhanced in LSM-SDC with respect to LSM ($\approx 100$ fold enhancement) and to LSM-YSZ. Besides the activation energy $E_a$ is decreased by more than 1 eV ($\approx 2.9$ eV for stoichiometric LSM and $\approx 1.8$ eV for LSM-SDC), becoming comparable to LSCo ($E_a \approx 1.5$ eV). Please note that, even though much lower values are expected for porous Co-based bulk electrodes[49], here the relevant comparison is between dense thin films with potential application as functional layers. In Fig. 3b, also the $k$ value obtained by the FEM simulation of the APT oxygen isotope profiles is reported (red star), demonstrating the very good agreement with the impedance data. With knowledge on the elementary processes of oxygen incorporation in LSM (atomic adsorbate ($O^-_{ad}$) incorporation-limited ORR process)[47,50], it is also possible to provide a mechanistic interpretation of the synergistic LSM-SDC incorporation mechanism by analyzing the oxygen partial pressure ($pO_2$) dependence of $k^q$. Figure 3c shows that in the low-$pO_2$ regime, both the single phase material and the nanocomposite follow the expected $\approx 1/2$ slope resulting from the

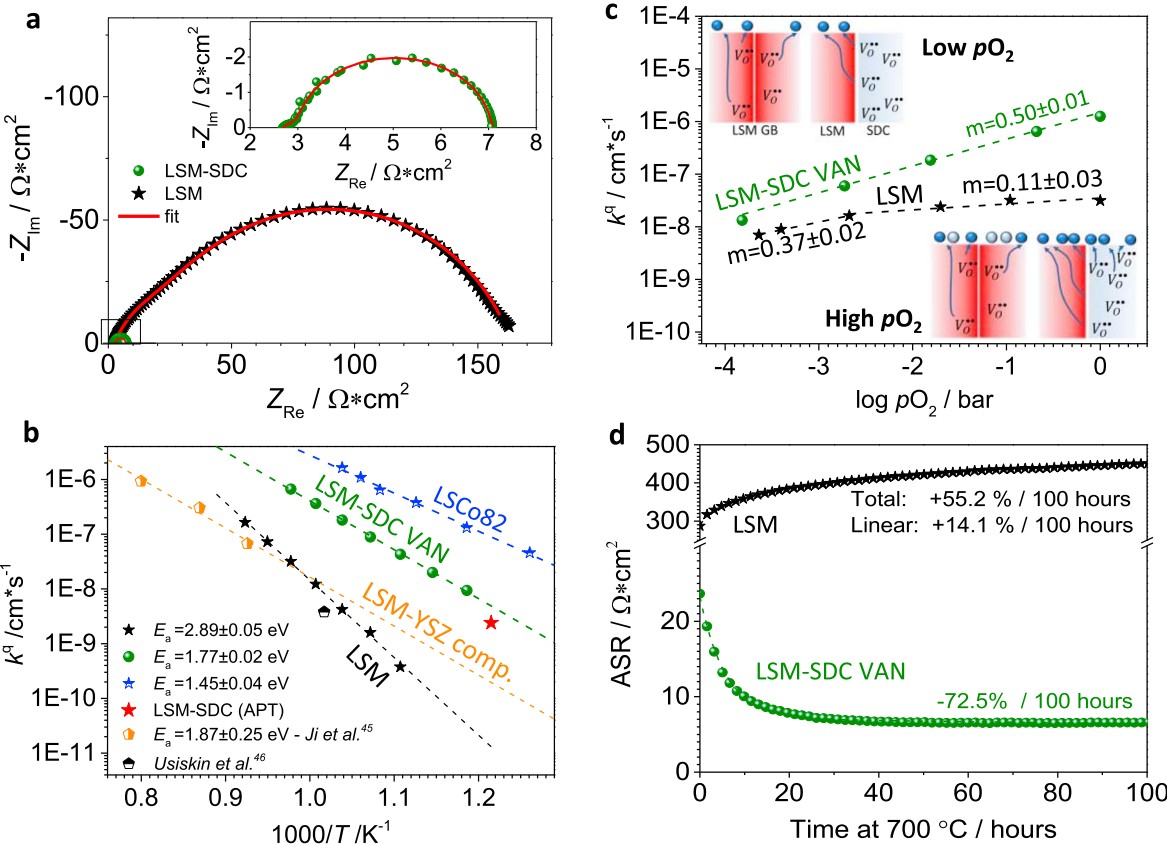

**Fig. 3 Electrochemical performance of LSM (black stars) and LSM-SDC (green bullets) nanocomposites. a** Representative initial EIS Nyquist plots measured at 750 °C (air atmosphere) in an out-of-plane configuration and using a porous Ag layer as a low impedance counter-electrode for LSM and LSM-SDC. The inset shows the LSM-SDC impedance arc on a smaller scale. **b** Surface exchange coefficient $k^q$ as a function of inverse temperature, as compared to stoichiometric LSM (from our labs and from Ref. [46]), state-of-the-art LSM-YSZ dense composites (from Ref. [45]) and to state-of-the-art La$_{0.8}$Sr$_{0.2}$CoO$_3$ (our lab). The red star represents the value resulting from the FEM simulation of the APT $^{18}$O fraction profile. **c** $pO_2$ dependence for $k^q$. The slopes for LSM and LSM-SDC are indicated. **d** ASR vs time for LSM and LSM-SDC, measured during ageing treatment. The lines are intended as a guide to the eye.

final incorporation step (i.e., the encounter between $O_{ad}^-$ and a lattice oxygen vacancy ($V_O^{\bullet\bullet}$), leading to a predicted reaction rate $\mathcal{R}\,(pO_2) \propto \theta\,(O_{ad}^-) \propto pO_2^{\frac{1}{2}}$, where $\theta$ is the surface coverage) being limiting in both cases. Under such conditions, also the absolute $k^q$ values are similar between the single-phase material and nanocomposite, further corroborating this interpretation. However, at high $pO_2$, the limited availability of oxygen vacancies in LSM (which are mostly concentrated at the grain boundaries) governs the saturation in the kinetics of the incorporation process (i.e., sluggish $pO_2$-$k^q$ dependence), resulting in an excess of $O_{ad}^-$ at the film surface (cf. also Supplementary Fig. 11)[51]. Conversely, in the case of LSM-SDC, the supply of oxygen vacancies from SDC allows for readily incorporating oxygen also at high $pO_2$, as schematically depicted in the sketches in Fig. 3c. This further demonstrates the beneficial effect of the vertical phase alternation and maximized triple-phase boundary density in the VAN structure for engineering electrochemical functional layers. Notably, oxygen incorporation in our VANs may be facilitated by the extension of the active area for oxygen incorporation also to the SDC surface as a consequence of adsorbates "spill-over" from LSM to SDC[52]. Such considerations also allow explaining the observed strong reduction of the ORR activation energy, which passes from ≈2.89 eV for single phase LSM to ≈1.77 eV for LSM-SDC: The fluorite phase—in intimate contact with LSM via the VAN architecture—introduces a large amount of highly mobile oxygen vacancies in the system, through which surface

oxygen adsorbates can be readily incorporated. This is in agreement with past theoretical and experimental works on LSM bulk and grain boundaries, which assign to the approach of oxygen vacancies to the $O_{ad}^-$ the rate-determining step (RDS) for the ORR in LSM[53]. The LSM RDS is therefore alleviated in our VANs. Notably, these considerations are perfectly in line with previous works from the co-authors[43], in which a decrease in ORR activation energy was measured for B-site deficient LSM. There, it was concluded that lowering the Mn concentration generates a reduction of oxygen vacancy formation energy and a consequent enhancement of the ORR at the LSM grain boundaries. Interestingly, the final ORR activation energy value for strongly B-site deficient LSM is similar to the value of our VANs (≈1.8 eV)[54]. In both cases, introducing mobile oxygen vacancies in the system greatly enhances the ORR kinetics. In VANs, a better electrochemical performance is ensured by the higher mobility and concentration of oxygen vacancies in the SDC phase.

The formation of space-charge zones and especially of oxygen vacancies redistribution at the LSM-SDC interface could possibly act as an additional factor in favor of faster ORR kinetics as a consequence of oxygen vacancy accumulation at the LSM side of the interface[55]. This however is not expected to be a dominant effect since the space-charge width is very narrow in highly doped systems. In our case, for SDC and LSM nominal acceptor concentration is 20% at.; From here, one can estimate the Debye

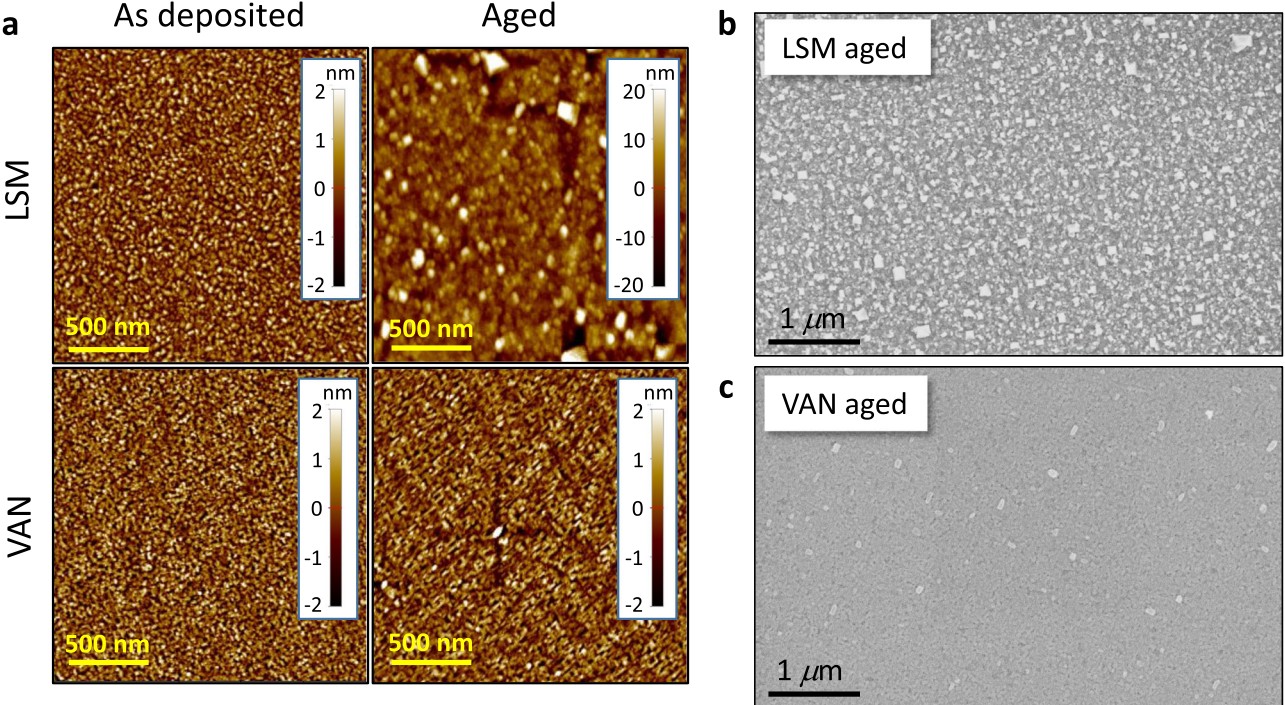

**Fig. 4 AFM and SEM top-view micrographs for LSM and LSM-SDC.** In (**a**), a full comparison between the surfaces of the two compounds in the as-grown and aged states is provided. The resulting root mean square roughness are: $R_{ms} = 0.8$ (7.1) and $R_{ms} = 0.7(1.0)$ nm for LSM-SDC and single phase LSM as grown (after ageing), respectively. In (**b** –LSM) and (**c**—VAN), lower magnification SEM for the surfaces after thermal treatment (In-Lens detector – 3 kV acceleration voltage).

length to be $\approx 1$ nm[56], i.e., at least one order of magnitude lower than the pillar width.

As far as the in-plane current percolation is concerned, although a perfect VAN structure should not, in principle, allow for electronic current transport, one can imagine that the (yet typically negligible) electronic conductivity of the SDC under working conditions may be sufficient to ensure a certain in-plane percolation owing to the very small pillar width ($\approx 10$ nm). Current collection for the electrochemical experiments was in any case ensured by the utilization of porous Au paste (see also experimental section), which allows the electrode to be partially in contact with the gas atmosphere while offering a closely spaced percolating path thus limiting in-plane potential drops. Such an approach has been extensively used in the past for dense and porous thin film electrodes[57].

Lastly, in Fig. 3d we report the results on the thermal stability of the SDC-LSM nanocomposites, evaluated by carrying out a 100 h treatment at 700 °C in air, while in-situ measuring the polarization resistance. It is well-known that, under such conditions, LSM and doped perovskite materials in general undergo changes in the surface composition as a consequence of dopant migration from the bulk, causing the formation of insulating Sr-based secondary phases[58,59]. In our case, single-phase LSM undergoes the expected degradation, which is reflected in a 50% ASR increase in 100 h. Surprisingly, the LSM-SDC nanocomposites are characterized by stable thermal behavior with virtually no degradation over the time span under consideration: Even a 70% decrease of the ASR is observed before stabilization (Please refer to Supplementary Fig. 12 for the time evolution of $D^q$ and $k^q$ for LSM and VANs).

**Top-view microscopy highlights no dopant segregation in VANs**. In order to rationalize the observed long-term electrochemical stability, we first carried out complementary

microstructural characterization by atomic force microscopy (AFM) and scanning electron microscopy (SEM). The results are shown in Fig. 4. In Fig. 4a, the comparison between the AFM micrographs from LSM and LSM-SDC before and after thermal treatment (700 °C, 100 h) highlights that no surface evolution is apparent for LSM-SDC, while secondary phase outgrows are formed on the LSM surface. These are ascribed to the well-known formation of Sr-based precipitates as a consequence of massive dopant migration which is typical of doped MIECs, and which is responsible for the severe degradation of the surface reactivity owing to a reduced availability of active sites[60]. The top-view SEM images for the materials after ageing (LSM and LSM-SDC in panels b and c, respectively), provide information at lower-magnification and highlight the profound difference between the two surfaces: Namely, deleterious Sr segregation is fully hindered in LSM-SDC. Please note that the bulk thermal stability of VANs had been already been proven by TEM (cf. Fig. 1).

**Surface chemical analysis by LEIS**. Complementarily to AFM and SEM imaging, the stability of VANs surface upon thermal ageing has been quantitatively assessed using state-of-the-art low energy ion scattering (LEIS). LEIS is capable of probing the elemental composition of the very outer atomic surface layer of the material and is therefore well-suited for analyzing surface chemical composition with the highest sensitivity[42]. The LEIS spectra for VAN as-grown and after thermal ageing (100 h at 700 °C) are reported in Fig. 5a, b, respectively. The stoichiometric ratios [Sr]/[La] can be obtained from the integration of the peak areas. It becomes evident that not just Sr has a very low tendency to segregate in the VAN (the final ratio [Sr]/[La] = 0.36 ± 0.06 is close the bulk value after the ageing heat treatment—Fig. 5b) but also that the Sr fraction is remarkably decreased in the final state with respect to the material as-grown ([Sr]/[La] = 0.86 ± 0.05— Fig. 5a). Such a reduction of the Sr content at the surface not only

confirms the high thermal stability of the VANs, but also explains the observed ASR evolution over time, which decreases during the first ≈40 h and is constant afterwards—cf. Fig. 3d. This indicates that Sr-La substitution is thermodynamically unfavorable at the VANs surface. (Please note the dominant Sr-termination in the as-deposited state is expected to lead to slower ORR kinetics even

in the absence of precipitates[61].) The initial high Sr content for the as-grown material also reveals that the thermodynamic conditions (high $T$ and low $pO_2$) and fast kinetics of the film synthesis process (PLD) are very far from the final equilibrium in air, while a bulk-like equilibrium situation is approached upon annealing in the VANs structure. Such a behavior is in contrast with the common behavior of MIEC materials showing thermally-induced Sr segregation and represents an outstanding improvement with respect to state-of-the-art MIEC electrodes.

**DFT: effects of local high entropy on lattice stabilization.** We performed theoretical calculations based on spin polarized Density Functional Theory (DFT—see further details in Supplementary Note 5 and elsewhere in ref. [62]) of a cubic perovskite supercell containing 40 atoms (Fig. 6a) with the aim of obtaining further insights into the observed thermal stability of the LSM-SDC VAN. Our LSM columnar structures, as highlighted experimentally, are characterized by high levels of cation intermixing (Sm and Ce) and by isotropic compressive strain (cf. Fig. 1). By carrying out energy minimization (Fig. 6b), we found that the introduction of Sm cations into the LSM model ($La_{0.75}Sr_{0.125}Sm_{0.125}MnO_3$) leads to a pronounced cell contraction of around 0.3%. Ce doping, even to quite high levels, has a smaller effect upon the lattice parameter, yet a reduction of a cell parameter is predicted with a drop of ~0.1% for a composition of $La_{0.625}Sr_{0.125}Ce_{0.25}MnO_3$. Therefore, one can first conclude that the observed isotropic compressive strain of LSM in the VAN structure (cf. Fig. 1) is a consequence of the chemical incorporation of Sm from the SDC layers (with an additional but smaller influence of Ce at higher dopant contents)[63], ruling out an important role of off-plane epitaxial elastic strain possibly deriving from the observed lattice match epitaxy (cf. Fig. 1e). Taking this into account, we directly evaluated the cation interdiffusion as a possible source of the Sr segregation suppression. As a consequence of the observed spontaneous cationic intermixing, here nominal LSM is, as a matter of fact, a

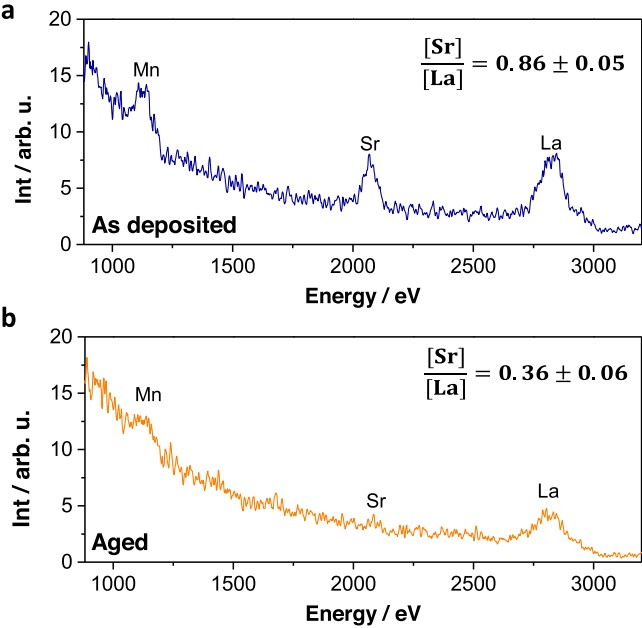

**Fig. 5 LEIS spectra for LSM-SDC VAN.** Analysis of surface as deposited (**a**) and after thermal treatment (100 h, 700 °C—**b**). The stoichiometric ratio is obtained from integration of the peak areas. Absolute differences in peak intensities should be ascribed to the specific experimental conditions.

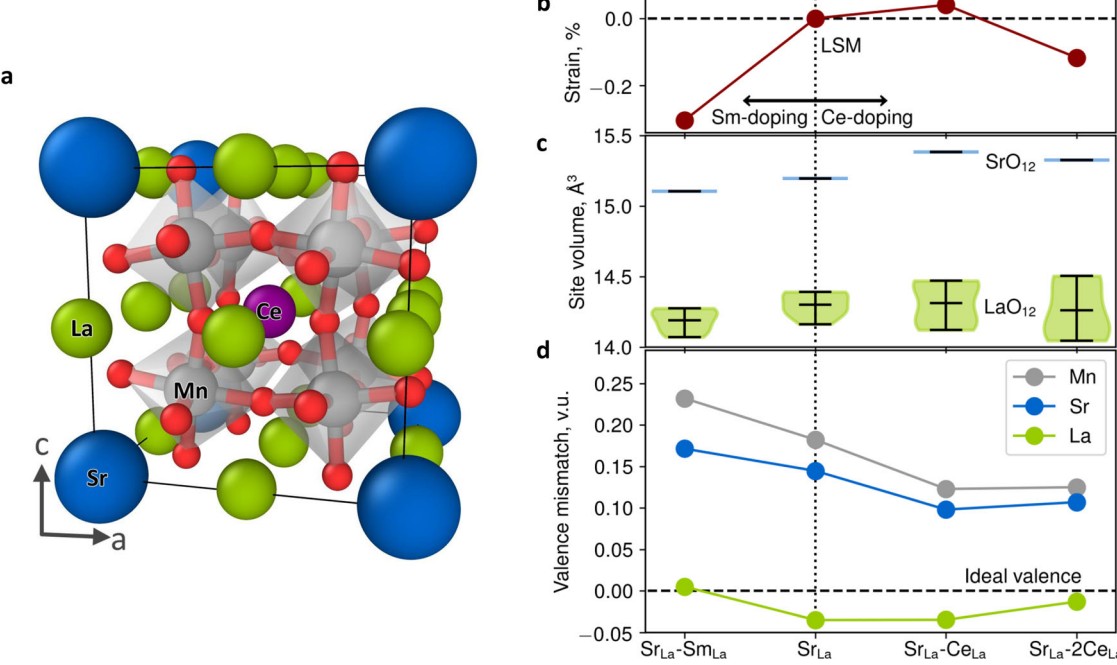

**Fig. 6 DFT calculations for high-entropy LSM. a** Example of a 40-atom periodic supercell used in DFT calculations showing relaxed positions of the Mn (gray), Sr (blue), La (green), O (red) and Ce (purple) ion and (**b**) calculated values of the lattice strain due to chemical doping. **c** box plot showing the mean and distribution of site volumes around the Sr (blue) and La (green) sites. **d** bond valence mismatch at the ion positions for the different cation dopants, where for example $Sr_{La}$-$Sm_{La}$ corresponds to a stoichiometry of $La_{0.75}Sr_{0.125}Sm_{0.125}MnO_3$.

six element high entropy oxide having $S_{config,VAN} = 1.21\,R$ ($S_{config} = -R\sum_{i=1}^{N} x_i\, ln x_i$ where $x_i$ is the atomic fraction in the unit formula as resulting from Table 1). The same calculation for $La_{0.8}Sr_{0.2}MnO_3$ yields a much lower value ($S_{config,LSM} = 0.56\,R$). Entropy-induced lattice stabilization has been recently reported for bulk LSM;[64] Here, we directly evaluate the effect of homogeneously dispersed multiple atom species in the perovskite A-site on the local strain field around the Sr atom, i.e., on the elastic driving force for atom surface segregation[60]. To this end, we considered the Sr (La)-O polyhedral volume (Fig. 6c) and the bond valence sum at each lattice site (Fig. 6d) for different cation distributions. In bulk LSM with a composition of $La_{0.875}Sr_{0.125}MnO_3$, we estimate that the Sr-ion is significantly overbonded with a valence mismatch of ~0.15 valence units (v.u.), i.e., the surrounding oxygen ions are closer than required by the formal valence state of the Sr ion, while La ions are slightly underbonded, ~−0.04 v.u. This opposite valence mismatch is the likely driving force for the exchange of overbonded Sr ions with underbonded La ions on the surface (in combination with electrostatic effects). This is conceptually similar to the Sr ion having a larger ionic radius than the La lattice ion it replaces[59]. Such an instability is strongly alleviated by cationic intermixing (Fig. 6c, d): Particularly, the introduction of Ce leads to an increase in the Sr-O site volume, in turn reducing the Sr-O overbonding relative to the La-O polyhedra and thereby decreasing the elastic driving force for Sr surface segregation. In other words, the increase of configurational entropy dictates the enhancement of the LSM stability by triggering a local lattice distortion. Besides, in agreement with recent theoretical works on high entropy oxides[65], one may consider also a stabilizing effect of the interfaces, i.e., the contribution of surface energy to the final free energy, which is maximized here by the very high interface density. Critically, the LSM in the VAN under consideration is able to accept Ce concentrations well beyond the reported solubility limits of Ce in single phase LSM (2% mol)[39], which is crucial for the observed effects and which further demonstrates the stability of the structure. Within the experimental resolution of the techniques, no detectable elemental clustering can be observed by HR-TEM (cf. Fig. 1) or by APT (cf. Fig. 2 and Supplementary Fig. 3), confirming the single solid-solution nature of our highly doped LSM in VANs. Unlike previously reported studies on entropy-stabilized LSM[64], in which no increase in the electrochemical properties was reported, the relevance of our VAN strategy lies in combining high thermal stability—owing to local high-entropy in the LSM phase—and strongly enhanced ORR kinetics due to the intimate phase alternation and off-plane geometry. Because the VAN structure is self-assembled, both the composition and interfaces which form at the growth temperature are the stable ones, i.e., they should not change with further annealing under conditions similar to the growth conditions as they are the energetically preferred. Please note that the TEM investigation presented in Fig. 1, showing very well-defined columnar structure and long-range order, was carried out after thermal ageing (100 h at 700 °C), confirming the structural stability of our VANs. This is very different to the case of artificially- grown superlattice structures, where the artificially-formed interfaces are unlikely to be the equilibrium ones (i.e., coherence is lost above a critical thickness and chemical instabilities may occur during operation).

In summary, we were able to combine, at the nanoscale, the excellent oxygen exchange properties of LSM with the good diffusivity of SDC to fabricate nearly ideal electrode functional layers for SOCs by design. This was achieved using VANs. Most importantly we highlighted that, by the spontaneous formation of a heavily co-doped manganite phase in the VAN (high entropy oxide), a stable compound is formed which hinders Sr-segregation to the surface. This resulted in an outstanding long-term durability at typical operational temperatures of solid oxide electrochemical cells.

The well-defined geometry, together with the intimate phase alternation and the long-range interface coherence, makes the structure ideal for directly probing structural and functional properties in great detail. Evidence of the presence of a fast oxygen diffusion pathway at the nanoscale along the SDC phase was presented here using isotopic exchange depth profiling based on APT. By this approach we proved that VAN nanocomposites can solve current kinetic limitations of existing electrodes by a synergistic combination of the functionalities of their individual constituents.

Overall, the described VAN architecture is an easy-to-implement, superior strategy for oxide electrochemistry resulting in long-term thermal stability and driving fast oxygen reduction kinetics. Our work represents a decisive step forward in the implementation of advanced thin film technology for high performance electrochemical energy conversion devices and opens up a path for the development of next-generation functional oxides.

## Methods

**Film fabrication.** LSM-SDC, LSM and LSCo thin films were deposited in a large-area pulsed laser deposition chamber (PVD Systems—PLD 5000) equipped with a 248 nm KrF excimer laser (Lambda Physics—COMPex PRO 205) on YSZ (100) substrate (Crystec GmbH) under the following conditions: temperature 800 °C (600 °C for LSCo), oxygen pressure 0.007 mbar, target-substrate distance 90 mm, laser fluency $1.1 \approx J \cdot cm^{-2}$, laser frequency 2 Hz for LSM-SDC, 10 Hz for LSM and LSCo. A thin (≈5 nm) buffer layer of $Ce_{0.8}Gd_{0.2}O_2$, fabricated under the same conditions, was deposited between LSCo and YSZ in order to avoid the formation of secondary phases. Since z-resolved $^{18}O$ labelling experiments are only feasible for films grown on an inert substrate, a perovskite $SrTiO_3$ substrate (100—Crystec GmbH) was employed for the sample that underwent APT analysis.

LSM-SDC films were fabricated using home-made targets obtained by mixing commercial powders of $La_{0.8}Sr_{0.2}MnO_3$ and $Ce_{0.8}Sm_{0.2}O_2$ (1:1 wt. %—Kceracell) via ball milling in ethanol solution. The dried powder mix was uniaxially pressed (7 MPa, 30 s) to form a pellet (diameter ≈ 1 inch). Sintering was carried out at 1300 °C for 4 h (heating and cooling ramps ≈5 °C/min). A commercial target was used for LSM and LSCo.

**Microscopy and XRD.** TEM and HAADF STEM imaging, EDS mapping and line scans were acquired by FEI Talos F200X. The TEM samples were prepared using conventional mechanical procedures including polishing, dimpling, and ion milling (PIPS 691). Films were analyzed by XRD by using a lab diffractometer with 4-angle goniometer (MRD X'Pert Pro from Malvern-Panalytical) and Cu Kα tube and 2 × Ge(110) monochromator. SEM and AFM characterization were performed in a Zeiss Auriga and a Park System, respectively.

**Atom probe tomography.** Atom probe tomography (Cameca LEAP 4000X Si) was performed on focused ion beam prepared specimens (FEI Helios NanoLab 600i) that were mounted on TEM grids and hardware that allowed for TEM imaging (FEI Talos F200X) and analysis of the APT specimens[66,67]. The 355 nm laser-pulsed APT analysis was performed at 57.3 K using either a 0.8 or 24 pJ laser energy and 500 kHz pulse rate (cf. Supplementary Table 1). The flight path length was 90 mm and the ion detection rate was set to 3.5 ions per 1000 pulses, resulting in a bias range of 2200–5000 V during the data collection. Reconstructions were generated in Cameca's IVAS 3.6.14 software using the TEM images of the specimens as a guide for setting the reconstruction parameters. Atomic volumes of the species were adjusted based on the unit cell parameters of the oxides.

**Oxygen tracer annealing.** $^{18}O$ tracer annealing was carried out at a temperature of 550 °C for 6000 s in an atmosphere of 200 mbar of enriched oxygen (90% $^{18}O$ concentration). Prior to the exchange, an annealing (12 h) in natural oxygen was carried out.

**Low energy ion scattering.** To explore the uppermost sample surface chemical composition, a Qtac100 LEIS instrument (ION-TOF GmbH) was employed. Neon primary ion beam was chosen for its higher mass resolution with respect to the elements of LSM-SDC composite analyzed in this study. $Ne^+$ 5 keV ion beam was rastered over a sample area of 500 µm × 500 µm. The energy range was maintained in 850–3600 eV. The $Ne^+$ dose was kept around $5 \times 10^{14}$ ions/cm$^2$ for each analysis. Cerium mean isotope LEIS peak overlaps with the lanthanum one. For this reason, the cerium higher mass isotope was also fitted to separate in this way cerium from lanthanum contribution.

**Electrochemical impedance spectroscopy**. The electrochemical characterization of the films was carried out using a Novocontrol impedance spectrometer using a frequency range $10^6-0.1$ Hz and a voltage amplitude of 0.05 V. A low-impedance Ag counter-electrode was applied by brushing Ag conductive paste (Sigma Aldrich), while a porous Au paste was applied on top of the films to ensure a homogeneous current distribution, limiting the in-plane potential drops while leaving a large fraction of the surface exposed to air. A fine gold mesh was used as a current collector on both sides. Measurements were carried out in a ProboStat test station (NorECs) placed inside a vertical furnace and using synthetic air atmosphere.

**FEM and DFT simulations**. FEM simulations were performed by COMSOL Multiphysics using the *Transport of diluted species* module. DFT theoretical calculations have been based on spin polarized DFT using plane wave CASTEP code[68]. These calculations used the Perdew, Burke, and Ernzerhof exchange correlation functional for solids (PBESol) in combination with ultrasoft pseudopotentials with a plane wave expansion to 600 eV and a Monkorst-Pack grid[69] spacing no >0.0404 Å$^{-1}$[70]. The presence of Mn within the LSM structure may cause a significant miscalculation of the exchange energy and so we use the DFT $+ U$ correction with a $U$-$J$ value of 4.0 eV on the Mn ions. Although largely empirically derived this value has been validated against more complete hybrid Hartree-Fock DFT calculations for the LSM structure and also in other Mn oxides[62]. An initial ordering of cations was chosen for study and the ion positions and cell parameters were relaxed according to a BFGS minimization. The initial spin polarization of the Mn ions was taken to be ferromagnetic and were free to refine during the progression of the minimization.

## Data availability

The APT, impedance spectroscopy. AFM, SEM and LEIS data that support the findings of this study are available in Zenodo (link) with the identifier "https://doi.org/10.5281/zenodo.4544312".

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

## Acknowledgements
J.S. acknowledges the support of ICN2 (funded by the CERCA programme/Generalitat de Catalunya and by the Severo Ochoa programme SEV-2017-0706) for the XRD measurements. M.A. acknowledges the support from the Feodor Lynen Research Fellowship Program of the Alexander von Humboldt Foundation and the Isaac Newton Trust, 17.25(a). M.A. and J.D. acknowledge the support from the EPSRC Centre of Advanced Materials for Integrated Energy Systems (CAM-IES) under EP/P007767/1. J.D. also acknowledge support from EPSRC grants EP/N004272/1, EP/T012218/1, the Royal Academy of Engineering- CIET1819_24, ERC POC grant 779444, Portapower. X.W. and H.W. acknowledge the funding support from the U.S. National Science Foundation for the TEM effort at Purdue University (DMR-1565822 and DMR-2016453). This project has received funding from the European Union's Horizon 2020 research and innovation program under grant agreement No 824072 (HARVESTORE), No 681146 (ULTRA-SOFC) and No 101017709 (EPISTORE) and was supported by an STSM Grant from the COST Action MP1308: Towards Oxide-Based Electronics (TO-BE), supported by COST (European Cooperation in Science and Technology). The authors thank A. Aguadero for the fruitful discussions and A. Kuzyk for support on graphics.

## Author contributions
Experiments were conceived by F.B., F.C., M.A., J.D., A.M., A.T. Sample preparation was carried out by F.B., M. A., electrochemical characterization by F.B. APT was performed by D.D., FEM simulations by F.C. DFT calculations were done by D.P. and A.Ch. XRD analysis was carried out by J.S., TEM analysis by X.W. under the supervision of H.W. A. Ca. performed the oxygen exchange treatment. F.B. designed the research plan and prepared the initial draft. All the authors contributed to the interpretation and discussion of the experimental data and to editing the initial draft.

## Competing interests
The authors declare no competing interests.
