## [Peer Review File · Nature Communications]

Reviewer #1 (Remarks to the Author):

The authors report a comprehensive experimental study on an ordered nanostructure composed of two phases, one being a good electric conductor (LSM) and the other being a good oxygen ion conductor (SDC). The nanocomposite acts as a dual-phase mixed-ionic-electronic-conductor (MIEC) and could be utilized as an oxygen electrode, e.g. in a solid oxide fuel cell or in an electrolyzer cell based on an oxygen ion conducting electrolyte, e.g. YSZ. The experiments seem to be done with great care, the conclusions are sound, and part of them are supported by DFT calculations. In summary, the manuscript contains sufficiently new data and new scientific insight to be published in Nature Communications.

I have, however, a few comments on how to improve the manuscript, but also some questions that need to be answered prior to a recommendation for publication.

1. The title is too unspecific. What are the “long-term energy applications”? There may be hundreds of energy applications coming to the mind of the reader. But then, in the text it turns out that the real application would be an oxygen-electrode in a Solid Oxide Cell (SOC). Of course, that is already something worthwhile to be improved, and worldwide many groups work on that topic, but it should become clear from the beginning.
2. The same applies to the abstract, it is too unspecific, and there are too many general statements. What is a “novel class of superior energy materials”? What are the “solid state energy devices”? After reading the abstract it remains completely unclear where the journey will go.
3. In the Introduction, the field of “dual phase membranes”, e.g. for oxygen separation from air, should be mentioned and referenced. The approach in this manuscript is a continuation of the old idea of a dual phase membrane, composed of an oxygen ion conductor and an electronic conductor.
4. P3: “...high electronic current density or high oxygen chemical potential...”. Probably it should read “...high oxygen chemical potential gradients...”?
5. P3: “ORR” should be explained.
6. P4: The first 14 lines in the chapter “Results and discussion” do not fit to the heading, they are a kind of literature review.
7. P5: Third line from the bottom: “...inset of panel d” should probably read “inset of panel e”.
8. P8, Table 1: Are there composition data for pure LSM and pure SDC using the same analysis method? These data would help to evaluate the reported compositions.
9. P8: “providing an unprecedented analysisunder controlled conditions of temperature and time.34,35”. The use of the word “unprecedented” is always questionable, and why references 34,35? It would be helpful to have a short explanation what was done in 34,35.
10. P9: “after the incorporation through the active surface.” It would be helpful to indicate where in Fig. 2 the active surface is located.
11. P9: “diffusion coefficient of LSM” should read “diffusion coefficient of oxygen in LSM”.
12. P9: “reported for related LSM-YSZ dense composites”. Only on p9 dense composites are mentioned for the first time. There are many publications on dense dual phase membranes, e.g. for oxygen separation from air. This big research field should have been mentioned already in the beginning (see point 4).
13. P18: “... i.e. they should not change with further annealing under conditions similar...” Are there experimental data supporting this expectation or hope?

Reviewer #2 (Remarks to the Author):

The manuscript 'A heavily substituted manganite in an ordered nanocomposite for long-term energy applications' by Baiutti et al. reports on structural properties and oxygen exchange kinetics of vertically aligned nanocomposites (VAN), combining LaSrMnO₃ and Sm-doped CeO₂, and showing enhanced long-term stability.

The authors achieved impressive nanostructured epitaxial composites on the active substrate YSZ and present a great collection of state-of-the-art methodology to characterize the ion transport and their kinetics on the nanoscale. Most notably, a detailed atom probe analysis of the nanocomposite chemistry is provided, complemented by chemical analysis via LEIS and electrochemical impedance spectroscopy. I therefore believe this study can be published in Nature Communication, but at the same time I would like to suggest to significantly refocus toward the methodological novelty presented.

I fully acknowledge the material synthesis and property analysis approach. However, it is not fully clear to me in how far this study revealed novel mechanistic insights, which are mentioned in the introduction and conclusion. The authors explain and demonstrate that the observed enhancement of oxygen exchange kinetics is mainly achieved by diffusion in the SDC, while the VAN structure seems to reflect a (very smart) way to increase the amount of triple phase boundaries at the surface. As far as I can judge, however, there is no demonstration of specific 'oxygen incorporation pathways' e.g. along the interfaces or similar. Also the stabilization of the LSM is in the end explained by chemical doping, and hence a bulk property of LSM. The mechanism leading to intermixing at the interfaces, and how it can be controlled is not addressed in detail. In terms of atomistic understanding, I am missing a comprehensive study where the number of interfaces / density of nanopillars is systematically varied.

I therefore suggest to focus more on the methodological novelty, while diminishing the interface engineering aspect a little bit, before this study should be published.

In addition to these general comments, I list below more detailed comments to the authors which should be addressed:

- Can the authors comment what is the role of space charges and charge-transfer along these interfaces, reflecting a key property of interfaces? What is the effect of gradient in ionic defect concentrations such as oxygen vacancies (which is huge in the SDC and small in the LSM) on the oxygen transport/exchange?
- Cation intermixing is always difficult to be determined from microscopic studies. Can the authors comment on resolution limits for cation intermixing in TEM and APT? Particularly in APT, how does the locally varying electrical conductivity between LSM and SDC affect the experiment? I assume a field-distribution needs to be anticipated in order to reconstruct the sample?
- Can the authors comment on Ce ions occupying A-sites in the perovskite structure? Which is the expected valence state of Ce?
- It appears from the TEM analysis that the coherent and ordered structure of the embedded LSM

pillars disappears close to the interface to the YSZ substrate. Is this true?

- Moreover, can the authors elaborate what is the role of the highly ordered structure deep insight the nanocomposite? Is only the surface-near structure important?

- In a perfect VAN structure it is difficult for me to imagine how (electrical) current transport is achieved? Is the network of nanopillars interconnected to provide sufficient in-plane conductivity?

- The O18 decay profiles (mentioned on page 9) into the LSM are not really clear to see. I suggest plotting e.g. Mn-profile and O18 profile in one panel.

- FEM analysis: Are there any nanoscopic input parameters required that specifically describe the behavior of interface?

- Fig. 3a: it should be indicated if these data reflect initial EIS spectra or after aging.

- Derivation of k_q : For the readers convenience it would be nice to include a few details on how k_q was extracted from the EIS spectra.

- Can the authors comment why the activation energy reflecting the surface exchange kinetics is changing? Can one specify which limitation is lifted in the nanocomposite? Related to that, what is the meaning of the observed change in slope with pO_2 ? Does it indicate that at low oxygen pressure, oxygen is incorporated directly into the LSM, while at high pO_2 incorporation into the SDC is preferred?

- The LEIS analysis seems to indicate that during growth Sr segregates toward the surface, while this accumulation seems to disappear after aging. Is it clear where the extra Sr is going?

- In the LEIS data, can the authors comment why also the intensity of Mn is reducing after aging?

- The authors argue that Sr segregation is related to a total energy argument, which can be avoided/reduced by chemical doping. Does it hence take place independent of ambient oxygen pressure? Can the authors comment how this compares to segregation phenomena triggered by the (partial) Schottky equilibrium such as in the case of $SrTiO_3$, where Sr segregation is triggered primarily at high oxygen pressure in order to form Sr vacancy defects?

- Finally, the authors bring in a 'high-entropy'-argument at the end of their paper. Can the authors comment if in that case a solid solution would be preferred against the coherently ordered VANs in order to further increase entropic energy?

REVIEWER COMMENTS

Reviewer #1 (Remarks to the Author):

The authors report a comprehensive experimental study on an ordered nanostructure composed of two phases, one being a good electric conductor (LSM) and the other being a good oxygen ion conductor (SDC). The nanocomposite acts as a dual-phase mixed-ionic-electronic-conductor (MIEC) and could be utilized as an oxygen electrode, e.g. in a solid oxide fuel cell or in an electrolyzer cell based on an oxygen ion conducting electrolyte, e.g. YSZ. The experiments seem to be done with great care, the conclusions are sound, and part of them are supported by DFT calculations. In summary, the manuscript contains sufficiently new data and new scientific insight to be published in Nature Communications.

First of all, the authors would like to thank the reviewer for appreciating our work and for carefully evaluating the manuscript. The very detailed review which was provided allowed us to improve the manuscript. Please find in the following the point-to-point answers to the reviewer's comments.

I have, however, a few comments on how to improve the manuscript, but also some questions that need to be answered prior to a recommendation for publication.

1. The title is too unspecific. What are the “long-term energy applications”? There may be hundreds of energy applications coming to the mind of the reader. But then, in the text it turns out that the real application would be an oxygen-electrode in a Solid Oxide Cell (SOC). Of course, that is already something worthwhile to be improved, and worldwide many groups work on that topic, but it should become clear from the beginning.

Following the reviewer's suggestion, we modified the title by introducing the application field “solid oxide cells”. We also propose to replace the expression “heavily substituted” with the more appropriate “high entropy”. The new title is the following: “A high-entropy manganite in an ordered nanocomposite for long-term application in solid oxide cells”.

2. The same applies to the abstract, it is too unspecific, and there are too many general statements. What is a “novel class of superior energy materials”? What are the “solid state energy devices”? After reading the abstract it remains completely unclear where the journey will go.

The abstract has been modified accordingly. “Novel class of superior energy materials” has been replaced by “novel class of superior functional materials with enhanced electrochemical properties” and “solid state energy devices” has been replaced by “solid state energy conversion devices such as solid oxide cells”.

3. In the Introduction, the field of “dual phase membranes”, e.g. for oxygen separation from air, should be mentioned and referenced. The approach in this manuscript is a continuation of the old idea of a dual phase membrane, composed of an oxygen ion conductor and an electronic conductor.

The sentence: “The structure combines an electronic conductor with a good electrocatalytic performance such as $\text{La}_{0.8}\text{Sr}_{0.2}\text{MnO}_3$ (LSM) with a fast ionic conductor, namely $\text{Ce}_{0.8}\text{Sm}_{0.2}\text{O}_2$ (SDC),²¹ in a

vertically aligned, nanoscale phase alternated, fashion (vertically aligned nanostructures – VANs)²⁴” has been replaced by: “The structure combines an electronic conductor with a good electrocatalytic performance such as $\text{La}_{0.8}\text{Sr}_{0.2}\text{MnO}_3$ (LSM) with a fast ionic conductor, namely $\text{Ce}_{0.8}\text{Sm}_{0.2}\text{O}_2$ (SDC).²¹ MIEC-oxygen conductor nanocomposites find wide application as SOCs electrodes and have also been employed as dual phase membranes for oxygen separation owing to the ability of transporting both ionic and electronic species while maintaining structural stability.^{22–24} Unlike such “classical” approaches, in this study the structure is characterized by a vertically aligned, nanoscale phase alternated, fashion (vertically aligned nanostructures – VANs).²⁵ ”

The following references have been added to the new version, to explicitly refer to dual phase oxygen membranes:

B. Wang, J. Yi, L. Winnubst and C. Chen, Stability and oxygen permeation behavior of $\text{Ce}_{0.8}\text{Sm}_{0.2}\text{O}_{2-\delta}$ - $\text{La}_{0.8}\text{Sr}_{0.2}\text{CrO}_{3-\delta}$ composite membrane under large oxygen partial pressure gradients, *J. Memb. Sci.*, 2006, 286, 22–25.

W. Li, T. F. Tian, F. Y. Shi, Y. S. Wang and C. S. Chen, $\text{Ce}_{0.8}\text{Sm}_{0.2}\text{O}_{2-\delta}$ - $\text{La}_{0.8}\text{Sr}_{0.2}\text{MnO}_{3-\delta}$ dual-phase composite hollow fiber membrane for oxygen separation, *Ind. Eng. Chem. Res.*, 2009, 48, 5789–5793.

4. P3: “...high electronic current density or high oxygen chemical potential...”. Probably it should read “...high oxygen chemical potential gradients...”?

The sentence has been corrected as suggested by the reviewer.

5. P3: “ORR” should be explained.

The explanation of the acronym (oxygen reduction reaction) has been added to the new version of the manuscript.

6. P4: The first 14 lines in the chapter “Results and discussion” do not fit to the heading, they are a kind of literature review.

This part has been moved to the introduction.

7. P5: Third line from the bottom: “...inset of panel d” should probably read “inset of panel e”.

The letter has been corrected. We take the occasion to thank the reviewer for the very detailed review which has been provided.

8. P8, Table 1: Are there composition data for pure LSM and pure SDC using the same analysis method? These data would help to evaluate the reported compositions.

In the following table, we report the results on the measured atomic fractions for LSM and SDC, carried out by APT on reference single phase thin film samples for different laser energies. One can observe that two laser energy windows exist (<1 pJ and =24 pJ – highlighted in green in the table), for which the ionic

concentrations are more accurately quantified. Such conditions have been used for designing the experiment involving the analysis of the VANs. The table is now included in the manuscript as Supplementary Table 1.

The following sentence has been added to the main text: “In Supplementary Table 1, we also report the results on the quantification of atomic fractions for single phase LSM and SDC films that we performed in order to assess the accuracy of the APT method under different measurement (laser energy) conditions.”

La _{0.8} Sr _{0.2} MnO ₃ – measured at.%					
Laser (pJ)	La	Sr	Mn	O	RMS dev
0.5	0.17	0.05	0.23	0.54	0.07
1	0.18	0.05	0.23	0.53	0.08
2	0.19	0.05	0.24	0.52	0.10
4	0.20	0.05	0.24	0.50	0.11
7	0.21	0.06	0.25	0.49	0.13
10	0.22	0.06	0.24	0.48	0.14
14	0.22	0.05	0.24	0.49	0.13
18	0.21	0.05	0.23	0.50	0.12
24	0.19	0.04	0.23	0.54	0.08
30	0.16	0.01	0.15	0.68	0.10
nominal					
0.5	0.16	0.04	0.2	0.6	

Ce _{0.8} Sm _{0.2} O _{1.9} – measured at.%				
Laser (pJ)	Ce	Sm	O	RMS dev
0.8	0.29	0.08	0.63	0.03
1.5	0.29	0.08	0.63	0.03
3	0.29	0.08	0.63	0.03
6	0.30	0.08	0.62	0.04
12	0.30	0.08	0.62	0.05
18	0.31	0.09	0.61	0.06
24	0.31	0.09	0.60	0.06
30	0.31	0.09	0.60	0.07
nominal				
0.8	0.28	0.07	0.66	

9. P8: “providing an unprecedented analysisunder controlled conditions of temperature and time.^{34,35}”. The use of the word “unprecedented” is always questionable, and why references 34,35? It would be helpful to have a short explanation what was done in 34,35.

The sentence: “we employed APT for providing an unprecedented analysis of the the electrochemical functionalites” has been replaced by: “we employed APT as a novel technique for analysing the electrochemical functionalities”. The sentence “The sensitivity of APT towards oxygen isotopes has been demonstrated previously.^{36,37}” has been added in order to explain the content of the two references under consideration.

In the abstract, we replaced the expression “unprecedented thermal stability” with “suppressed dopant segregation”.

10. P9: “after the incorporation through the active surface.” It would be helpful to indicate where in Fig. 2 the active surface is located.

We modified the sentence accordingly, which now reads: “after the incorporation through the active surface (top of the APT tip in Fig. 2)”.

11. P9: “diffusion coefficient of LSM” should read “diffusion coefficient of oxygen in LSM”.

The correction has been implemented.

12. P9: “reported for related LSM-YSZ dense composites”. Only on p9 dense composites are mentioned for the first time. There are many publications on dense dual phase membranes, e.g. for oxygen separation from air. This big research field should have been mentioned already in the beginning (see point 4).

Dual phase membranes and related references are now included in the introduction of the manuscript as described in P4.

13. P18: “... i.e. they should not change with further annealing under conditions similar...” Are there experimental data supporting this expectation or hope?

We would like to draw the reviewer’s attention to the fact the TEM investigation presented in Fig. 1 was carried out after thermal ageing at 700 °C for 100 hrs, with the purpose of demonstrating the long term stability of the system. This is now stated more clearly in the manuscript at page 19, where the following sentence has been added: “Please note that the TEM investigation presented in Fig. 1, showing very well-defined columnar structure and long-range order, was carried out after thermal ageing (100 hrs at 700 °C), confirming the structural stability of our VANs.”

Reviewer #2 (Remarks to the Author):

The manuscript ‘A heavily substituted manganite in an ordered nanocomposite for long-term energy applications’ by Baiutti et al. reports on structural properties and oxygen exchange kinetics of vertically aligned nanocomposites (VAN), combining LaSrMnO₃ and Sm-doped CeO₂, and showing enhanced long-term stability.

The authors achieved impressive nanostructured epitaxial composites on the active substrate YSZ and present a great collection of state-of-the-art methodology to characterize the ion transport and their kinetics on the nanoscale. Most notably, a detailed atom probe analysis of the nanocomposite chemistry is provided, complemented by chemical analysis via LEIS and electrochemical impedance spectroscopy. I therefore believe this study can be published in Nature Communication, but at the same time I would like to suggest to significantly refocus toward the methodological novelty presented.

I fully acknowledge the material synthesis and property analysis approach. However, it is not fully clear to me in how far this study revealed novel mechanistic insights, which are mentioned in the introduction and conclusion. The authors explain and demonstrate that the observed enhancement of oxygen exchange kinetics is mainly achieved by diffusion in the SDC, while the VAN structure seems to reflect a (very smart) way to increase the amount of triple phase boundaries at the surface. As far as I can judge, however, there is no demonstration of specific ‘oxygen incorporation pathways’ e.g. along the interfaces

or similar. Also the stabilization of the LSM is in the end explained by chemical doping, and hence a bulk property of LSM. The mechanism leading to intermixing at the interfaces, and how it can be controlled is not addressed in detail. In terms of atomistic understanding, I am missing a comprehensive study where the number of interfaces / density of nanopillars is systematically varied.

I therefore suggest to focus more on the methodological novelty, while diminishing the interface engineering aspect a little bit, before this study should be published.

We would like to thank the reviewer for appreciating our work and to acknowledge its very careful review, which we assessed by thoroughly reviewing the manuscript. We believed that the changes we implemented in response to this general comment have indeed led to a shift of the focus towards the methodological novelties, as suggested. At the same time, with the point-to-point answers below, we strengthened aspects related to the interpretation of our data. Altogether, we believe that the new version of the manuscript has been greatly improved in terms of focus and clarity and we thank once again the reviewer for its valuable contribution.

In the following, we present the main changes that we carried out in response to the reviewer's comment.

In the abstract, the following changes have been carried out:

“Vertically aligned nanocomposites are characterized by a coherent, dense array of vertical interfaces, which allows for the extension of local effects to the whole volume of the material. Here, we use such a unique architecture to fabricate highly electrochemically active nanocomposites...” has been replaced by “Vertically aligned nanocomposites, which are characterized by a coherent, dense array of vertical interfaces, are here employed to fabricate highly electrochemically active nanocomposites...”

“Direct evidence of synergistic local effects for enhancing the electrochemical performance, stemming from the highly ordered phase alternation, is given here for the first time using atom-probe tomography combined with oxygen isotopic exchange” has been replaced by: “By a detailed analysis using complementary state-of-the-art techniques, which include novel atom-probe tomography combined with oxygen isotopic exchange, we assess the local structural and electrochemical functionalities and we allow direct observation of nanoscaled fast oxygen diffusion pathways.”

“Interface-induced cationic substitution, enabling lattice stabilization, is presented as the origin of the observed long-term stability. ” has been replaced by “Finite element method simulations ascribe the origin of the observed long-term stability to spontaneous cationic substitution enabling lattice stabilization.”

“These findings reveal a novel route for materials nano-engineering based on the coexistence between local disorder and long-range arrangement” has been replaced by “This work introduces a novel advanced method for the local analysis of mass transport phenomena and highlights the relevance of nano-engineering based on local disorder and long-range arrangement.”

In the introduction:

On page 3, we added: “This goes hand-in-hand with the development of novel techniques which are capable of capturing nanoscaled phenomena with higher accuracy.”

We removed: “The obtained ordered nanostructure is also ideal for fundamental investigations on oxygen reduction reaction (ORR) kinetics.”

We modified the sentence: “The obtained ordered nanostructure is also ideal for fundamental investigations on oxygen reduction reaction (ORR) kinetics”, which now reads: “The obtained ordered nanostructure is also ideal for implementing complementary state-of-the art and novel techniques for locally assessing structural and functional aspects.”

The expression: “Direct evidence of a synergistic mechanisms for oxygen incorporation and mass transport at the nanoscale has been obtained by...” has been replaced by: “Direct visualization of a nanoscaled fast mass transport pathway along the SDC matrix has been obtained by...” and later in the same paragraph “...achieving unique nm-resolution for describing the incorporation mechanism” has been replaced by: “achieving unique nm-resolution for describing the oxygen diffusion.”

In results and discussion:

On page 10: “direct observation of the oxygen kinetics” has been replaced by: “direct observation of the fast oxygen pathways”

In section conclusions:

The sentence: “Such exceptional lattice stability is extended to the whole thin film electrode due to the unique features of VANs, which are able to extend genuine interface effects to the whole volume” has been removed.

The sentence: “The off-plane geometry, together with the intimate phase alternation and the long-range interface coherence, makes the structure ideal for directly probing solid-gas reactions and oxygen transport” has been modified as follows: “The well-defined geometry, together with the intimate phase alternation and the long-range interface coherence, makes the structure ideal for directly probing structural and functional properties in great detail.”

In addition to these general comments, I list below more detailed comments to the authors which should be addressed:

- Can the authors comment what is the role of space charges and charge-transfer along these interfaces, reflecting a key property of interfaces? What is the effect of gradient in ionic defect concentrations such as oxygen vacancies (which is huge in the SDC and small in the LSM) on the oxygen transport/exchange?

The extent of the space-charge as well as of the charge-transfer zones is dependent on the doping level of the constituting materials (c.f. e.g. Gunkel et al., Phys. Rev. B 2016, 93, 245431). In our case, the SDC and LSM the nominal acceptor concentration is 20% at.; From here, one can estimate the Debye length to be ≈ 1 nm (c.f. Kim et al., Phys. Chem. Chem. Phys. 2016, 18, 3023), i.e. at least one order of magnitude lower than the pillar width. In such a situation, space-charge effects are not expected to be dominant. The reviewer raises an interesting question regarding the concentration gradient for oxygen at the interface, which indeed may possibly lead to oxygen vacancy migration from SDC to LSM (c.f. Baiutti et al., Nanoscale 2018, 10, 8712-8720) and in turn determine the formation of a vacancy depletion zone in SDC at the interface within the space-charge zones. Oxygen transport in the VAN

electrode is however expected to run along the pillars (following the electrochemical potential gradient) and thus remains largely unaffected by such blocking boundaries. Conversely, an increased concentration of oxygen vacancies in the space-charge zones of LSM could support an enhancement in the ORR kinetics. We, however, predominantly assign the observed fast ORR to a job-sharing mechanism – as described in the paper – owing again to the very low extent of the space-charge zones.

The following sentence has been added to the manuscript on page 15: “The formation of space-charge zone and especially of oxygen vacancies redistribution at the LSM-SDC interface could possibly act as an additional factor in favor of faster ORR kinetics as a consequence of oxygen vacancy accumulation at the LSM side of the interface.⁵³ This however is not expected to be a dominant effect since the space-charge width is very narrow in highly doped systems.”

The following reference has been added: F. Baiutti, G. Gregori, Y. E. Suyolcu, Y. Wang, G. Cristiani, W. Sigle, P. A. Van Aken, G. Logvenov and J. Maier, High-temperature superconductivity at the lanthanum cuprate/lanthanum-strontium nickelate interface, *Nanoscale*, 2018, 10, 8712–8720.

- Cation intermixing is always difficult to be determined from microscopic studies. Can the authors comment on resolution limits for cation intermixing in TEM and APT? Particularly in ATP, how does the locally varying electrical conductivity between LSM and SDC affect the experiment? I assume a field-distribution needs to be anticipated in order to reconstruct the sample?

The reviewer brings up a fair point about one of the challenges of APT analysis of multiphase materials – accounting for differences in evaporation fields. In the present case, the LSM phase has a lower evaporation field than the SDC phase so we might expect changes in the local radius of curvature near the interface of the phases to result in ions from the SDC phase close to the LSM interfaces to have trajectories that, when back-projected onto a presumed smoothly curved surface, result in apparent intermixing of the phases near the interfaces. Previous research has indicated that the extent of this overlap is generally up to about 2 nm around the interfaces. Those studies have also indicated that for regions larger than 2 nm, the “core” of the lower evaporation field phase that excludes the outer 2nm of material retains the accurate composition of that phase. We would also like to point out that the present version of the paper now also contains (Supplementary Table 1) results from the atomic fraction quantification for single phase LSM and SDC films, which were carried out as a test experiment before measuring the VAN sample. This experiment demonstrates that, for certain bias / laser energy conditions, the stoichiometry can be correctly quantified reasonably well for both phases. Supplementary Table 1 is reported also here for the reviewer’s convenience. In green, we highlight the final laser energy windows used in the experiment:

La _{0.8} Sr _{0.2} MnO ₃ – measured at.%					
Laser (pJ)	La	Sr	Mn	O	RMS dev
0.5	0.17	0.05	0.23	0.54	0.07
1	0.18	0.05	0.23	0.53	0.08
2	0.19	0.05	0.24	0.52	0.10
4	0.20	0.05	0.24	0.50	0.11
7	0.21	0.06	0.25	0.49	0.13
10	0.22	0.06	0.24	0.48	0.14
14	0.22	0.05	0.24	0.49	0.13
18	0.21	0.05	0.23	0.50	0.12
24	0.19	0.04	0.23	0.54	0.08
30	0.16	0.01	0.15	0.68	0.10
nominal					
0.5	0.16	0.04	0.2	0.6	

Ce _{0.8} Sm _{0.2} O _{1.9} – measured at.%				
Laser (pJ)	Ce	Sm	O	RMS dev
0.8	0.29	0.08	0.63	0.03
1.5	0.29	0.08	0.63	0.03
3	0.29	0.08	0.63	0.03
6	0.30	0.08	0.62	0.04
12	0.30	0.08	0.62	0.05
18	0.31	0.09	0.61	0.06
24	0.31	0.09	0.60	0.06
30	0.31	0.09	0.60	0.07
nominal				
0.8	0.28	0.07	0.66	

Based on the reviewer's comments, we have incorporated the above aspects, which were previously included as Supplementary Information, into the main text as follows:

The following text has been added to the main text on pages 8-9:

“One challenge in APT analysis of multiphase materials is that differences in the evaporation fields of the materials can result in changes in the local radius of curvature near the interfaces of the phases which produce ion trajectories that deviate from that presumed in generating the reconstruction of the analyzed volume.³⁶⁻³⁸”

The result is apparent intermixing of the higher evaporation field phase (SDC in this case) into the lower evaporation field phase (LSM). Previous research has indicated that the extent of this overlap is generally up to about 2 nm around the phase interfaces and that for regions larger than 2 nm, the “core” of the lower evaporation field phase that excludes the outer 2 nm of material retains the accurate composition of that phase.^{37,38”}

“In Supplementary Table 1, we also report the results on the quantification of atomic fractions for single phase LSM and SDC films that we performed in order to assess the accuracy of the APT method under different measurement (laser energy) conditions.”

The following references have been added:

M.K. Miller, M.G. Hetherington, Local magnification effects in the atom probe, *Surface Science* 246(1) (1991) 442-449.

F. Vurpillot, A. Bostel, D. Blavette, Trajectory overlaps and local magnification in three-dimensional atom probe, *Applied Physics Letters* 76(21) (2000) 3127-3129.

E.A. Marquis, F. Vurpillot, Chromatic Aberrations in the Field Evaporation Behavior of Small Precipitates, *Microscopy and Microanalysis* 14(6) (2008) 561-70.

Table 1 and related text have been modified to show the compositions measured in just the core of each phase's region.

As far as the TEM analysis is concerned, cation intermixing is indeed a challenging part of the microscopy study. However in TEM/STEM mode, the phase/composition analysis was conducted on a cross-sectional specimen without imaging reconstruction. Thus the study can be very complementary to the APT analysis. We have conducted local compositions and elemental analysis based on both the STEM and EDX mapping results and the phase separation between LSM and SDC columns are very obvious based on the STEM and EDX results. There are potential issues of column overlapping in the cross-sectional imaging mode causing challenges in revealing the interface. Thus we have focused on the edge on interfaces to reveal the sharpness of the two phases.

The following sentence has been added to the main text on page 8: "We note here that, while TEM imaging and spectroscopy result in a direct phase/composition analysis on large portions of a cross-sectional specimen, APT is able to provide complementary local information via a 3-dimensional reconstruction, with enhanced ability to quantitatively assess local elemental distributions."

- Can the authors comment on Ce ions occupying A-sites in the perovskite structure? Which is the expected valence state of Ce?

LSM doping by Ce was explicitly investigated in the past by the Irvine group (Konyshva et al., *J. Electrochem. Soc.* 2010, 157, B159-B165). Here, it was concluded that only a minor change of the electronic conductivity, and no change in the Mn oxidation state, occurs upon Ce doping. Therefore, we can assume that the main expected valence of Ce in LSM is 3+, i.e. Ce acts as an isovalent substitutional dopant for La. This conclusion is strengthened by comparing the ionic radii of La with Ce³⁺ (-1.47%) and Ce⁴⁺ (-16.17%).

The following sentence has been added to the manuscript on page 8: "As reported previously and as resulting from steric considerations, Ce is expected to be mostly present as an isovalent substitutional dopant for La in LSM, thus not directly affecting the electrochemical transport properties."^{37,38}

The following references have been added:

E. Konyshva, S. Francis, J. Irvine, Crystal Structure, Oxygen Nonstoichiometry, and Conductivity of Mixed Ionic–Electronic Conducting Perovskite Composites with CeO₂ *J. Electrochem. Soc.* 2010, 157, B159-B165

R.D. Shannon, Revised effective ionic radii and systematic studies of interatomic distances in halides and chalcogenides, *Acta Cryst.* 1976, A32, 751

- It appears from the TEM analysis that the coherent and ordered structure of the embedded LSM pillars disappears close to the interface to the YSZ substrate. Is this true?

It is indeed apparent that the nucleation of the two different phases occurs a few nm far from the film-substrate interface. Particularly, a Ce-rich phase has been highlighted by TEM. This can be explained in light of the structural and chemical affinity between the YSZ substrate and the fluorite SDC.

The following sentence has been added to the main text on page 6: "In Fig. 1c, it is also interesting to notice that the substrate-film interface is characterized by an even more pronounced intermixing and that no phase separation is apparent. We ascribed such a finding to the structural (lattice match) and chemical (wettability) properties of the fluorite YSZ substrate which favors the formation of a single, Ce-rich, phase during the first stages of film growth."

- Moreover, can the authors elaborate what is the role of the highly ordered structure deep inside the nanocomposite? Is only the surface-near structure important?

The highly ordered structure deep inside the nanocomposite ensures that out-of-plane ionic pathway has no tortuosity and is therefore maximized.

This is now stated clearly on page 13: "...the APT characterization of the oxygen kinetics highlighting fast vertical diffusion of oxygen through the SDC phase and is ascribed to the intrinsic fast ionic conductivity of the fluorite, together with the advantageous off-plane VAN geometry."

Also on page 10, "off-plane [mass transport]" has been added.

- In a perfect VAN structure it is difficult for me to imagine how (electrical) current transport is achieved? Is the network of nanopillars interconnected to provide sufficient in-plane conductivity?

Owing to the very small pillar width (≈ 10 nm), we believe that the (yet typically negligible) electronic conductivity of the SDC under working conditions may be sufficient to ensure a certain in-plane percolation. However, we agree with the reviewer that in-plane conductivity remains a bottleneck for ceramic electrodes in general and that the utilization of a metallic current collector is necessary in order to assess electrochemical properties. In our case, we utilized porous Au paste which allows the electrode to be (partially) in contact with the gas atmosphere while offering a closely spaced percolating path which limits in-plane potential drops. Such an approach has been extensively used in the past for dense and porous thin film electrodes (c.f. e.g. Wells et al., ACS Appl. Mater. Interfaces 2021, 13, 4117-4125 and Chen et al., J. Electroceram 2012, 28, 62-69).

This is now better described in the experimental section: "...a porous Au paste was applied on top of the films to ensure a homogeneous current distribution, limiting the in-plane potential drops while leaving a large fraction of the surface exposed to air."

Please note that the presence of the (ionically) conducting substrate prevented us from directly measuring the in-plane conductivity of our VAN system at high temperatures.

- The O18 decay profiles (mentioned on page 9) into the LSM are not really clear to see. I suggest plotting e.g. Mn-profile and O18 profile in one panel.

We modified Fig. 2. Panel (e) now also shows the Mn concentration profile as retrieved from the integration of the area marked in panel (d). The caption and the main text (page 10) have been modified

accordingly.

- FEM analysis: Are there any nanoscopic input parameters required that specifically describe the behavior of interface?

The FEM model describes the heterogeneous O18 diffusion inside the composite VAN solving Fick's second law under the boundary conditions shown in supplementary Fig. 6c. We did not consider any specific parameters for describing the interface, which is modeled as a continuum layer between the SDC and LSM pillars. The model used is similar to the FEM model developed by the authors in a previous work for describing fast grain boundary diffusion in $\text{La}_{0.8}\text{Sr}_{0.2}\text{Mn}_{1-x}\text{Co}_x\text{O}_{3-d}$ polycrystalline thin films (Saranya et al. Chem. Mater. 2018, 30, 16, 5621–5629), where two diffusivity coefficients were considered for the 1 nm width grain boundary (fast) and the grain interior (slow). In the FEM model presented in this work for the LSM/SDC VAN, fast diffusivity is considered in the SDC matrix and slow diffusivity in the LSM pillars.

The model shows that, due to the large difference of oxygen diffusivity in the SDC and LSM layers, the O18 incorporated on the surface rapidly fills the SDC matrix and then slowly diffuses into the LSM pillars. This complete filling of the SDC matrix prevents the observation of possible fast diffusion at the LSM/SDC interface (if present), which is anyway expected to take place in a very limited region, due to the high dopant concentration in the two components (see discussion above on space-charge effects). Regarding the in-plane oxygen transport from SDC to LSM, the experimental data do not suggest any singularity in the oxygen diffusivity at the interface, either because not present or because of a very limited effect on the behavior of the VAN thin film. For these reasons, the FEM model does not include any nanoscopic parameters for describing oxygen transport at the LSM/SDC interface, which is modeled as an abrupt change of diffusivity in the two materials.

We carried out the following modification of the main text: "...the measured diffusion profiles (integrated over the area between the dotted lines in Fig. 2d) were fitted using Finite Element Method (FEM) simulations on a modelled geometry (details on the modelling are given in Supplementary Note 3)" is now "the measured diffusion profiles (integrated over the area between the dotted lines in Fig. 2d) were fitted using Finite Element Method (FEM) simulations on a modelled 3D geometry consisting of LSM columns (with slow oxygen diffusivity) embedded in an SDC matrix (further details on the modelling are given in cf. Supplementary Note 3). The interface was modelled as a continuous layer."⁴⁴

The following reference has been added: "A. M. Saranya, A. Morata, D. Pla, M. Burriel, F. Chiabrera, I. Garbayo, A. Hornés, J. A. Kilner and A. Tarancón, Unveiling the Outstanding Oxygen Mass Transport Properties of Mn-Rich Perovskites in Grain Boundary-Dominated $\text{La}_{0.8}\text{Sr}_{0.2}(\text{Mn}_{1-x}\text{Co}_x)\text{O}_{3\pm\delta}$ Nanostructures, Chem. Mater., 2018, 30, 5621–5629."

- Fig. 3a: it should be indicated if these data reflect initial EIS spectra or after aging.

The caption of figure 3 has been modified accordingly: "a) Representative initial EIS Nyquist plots..."

- Derivation of k_q : For the readers convenience it would be nice to include a few details on how k_q was extracted from the EIS spectra.

The relation between R_s (surface resistance for the oxygen chemical reaction) and k^q has been added to the main text on page 13:

“By adjusting the EIS spectra of the cells with a physically meaningful equivalent circuit, the effective rate constant for the oxygen reduction reaction k^q can be obtained: $k^q = \frac{k_b \cdot T}{R_s \cdot c_{O_2} \cdot z_i^2 \cdot e^2}$, being R_s the surface resistance for the oxygen chemical reaction.”

- Can the authors comment why the activation energy reflecting the surface exchange kinetics is changing? Can one specify which limitation is lifted in the nanocomposite? Related to that, what is the meaning of the observed change in slope with pO₂? Does it indicate that at low oxygen pressure, oxygen is incorporated directly into the LSM, while at high pO₂ incorporation into the SDC is preferred?

According to pivotal works on LSM by J. Maier et coworkers (c.f. e.g. Matrikov et al., J. Phys. Chem. C 114, 3017-3017, 2010), the rate limiting step for oxygen incorporation has been identified as the encounter between an oxygen vacancy and an atomic oxygen adsorbate. More specifically, it is the (slow) vacancy approach that dictates the kinetics of the ORR; As a consequence and as abundantly shown in literature, there is a correspondence between the activation energy for the oxygen diffusion and for the surface ORR in LSM (c.f. Matrikov et al., Phys. Chem. Chem. Phys. 15, 5443-5471, 2013). In the case of the VANs under investigation, the SDC phase introduces in the system abundant and highly mobile oxygen vacancies through which oxygen adsorbates can be readily incorporated. Notably, these considerations are perfectly in line with previous works from the co-authors (Chiabrera et al., Adv. Mater. 2019, 31, 1805360 and Chiabrera et al., Solid State Ionics 299, 70-77, 2017), in which a decrease in ORR activation energy was measured for B-site deficient LSM. Here, it was concluded that lowering the Mn concentration generates a reduction of oxygen vacancy formation energy and a consequent enhancement of the ORR kinetics. Interestingly, the final ORR activation energy value for strongly B-site deficient LSM (≈ 1.6 eV – see figure R1, from F. Chiabrera, PhD thesis 2019) is in good agreement with the value of our VANs (≈ 1.8 eV): In both cases, introducing mobile oxygen vacancies in the system greatly enhances the ORR kinetics.

Figure R1. Activation energy for ORR and oxygen diffusion for LSM with different B/A ratio. From F. Chiabrera, Universidad Autonoma de Barcelona, PhD thesis 2019.

This has been included in the present version of the manuscript on page 14:

“Such considerations also allow explaining the observed strong reduction of the ORR activation energy, which passes from ≈ 2.89 eV for single phase LSM to ≈ 1.77 eV for LSM-SDC: The fluorite phase – in

intimate contact with LSM *via* the VAN architecture – introduces a large amount of highly mobile oxygen vacancies in the system, through which surface oxygen adsorbates can be incorporated. The typical LSM limiting step is therefore alleviated in our VANs. This is perfectly in line with past theoretical and experimental works on LSM bulk and grain boundaries, which assign the diffusion of oxygen vacancies as the rate-determining step for the ORR in LSM.^{41,50}

As far as the observed change in slope with pO₂ is concerned, the authors would like to point out that this has been measured for LSM, whereas the VAN maintain the ½ slope which is expected for the final incorporation step (i.e., the encounter between the atomic oxygen adsorbate and the lattice vacancy) being limiting. Fig 3c has been modified for clarity:

- The LEIS analysis seems to indicate that during growth Sr segregates toward the surface, while this accumulation seems to disappear after aging. Is it clear where the extra Sr is going?

According to the quantitative investigation which was carried out by APT (Fig. 2), we observed no Sr in the SDC despite non-negligible solubility (c.f. Anjaneya et al., *J. Alloys Compd.*, 598, 33-40, 2014). We conclude therefore that surface Sr gets reincorporated in the LSM lattice. The typical values for Sr diffusion coefficient ($\approx 10^{-15}$ cm²/s at 700 °C – cf. Ravella et al., *ChemistrySelect* 2017, 2, 5616-5623 and Kubicek et al., *Phys. Chem. Chem. Phys.* 2014, 16, 2715-2726) are consistent with the length scale of the VAN system.

The following sentence has been added to the manuscript: “Arguably, Sr excess at the VAN surface gets reincorporated in the LSM lattice during the thermal treatment”.

The following reference has been added: M. Kubicek, G. M. Rupp, S. Huber, A. Penn, A. K. Opitz, J. Bernardi, M. Stöger-Pollach, H. Hutter and J. Fleig, Cation diffusion in La_{0.6}Sr_{0.4}CoO_{3-δ} below 800 °C and its relevance for Sr segregation, Phys. Chem. Chem. Phys., 2014, 16, 2715–2726.

- In the LEIS data, can the authors comment why also the intensity of Mn is reducing after aging?

The absolute peak intensity of LEIS spectra is influenced by several parameters including analysis ion current, analysis area, impurity concentration on the surface, roughness. For this reason, even if the analysis conditions are the same, the species intensity of different analyses are not directly comparable but a normalization is necessary. In our case, we performed two LEIS analysis for each sample. The stoichiometric ratios indicated in Fig. 5 are the average of the results of the measurements. Notably, such ratios are independent on the normalization procedure. For the reviewer's convenience, we report here the absolute Mn concentration which we obtained for the different measurements. Normalization has been carried out by considering the sum of the peak areas for all the elements. One can observe that the Mn content remains constant during thermal ageing.

		Norm Mn	St. Dev. (1,2)
As grown	Analysis 1	0.30116	0.06657
	Analysis 2	0.39531	
Aged	Analysis 1	0.35956	0.05055
	Analysis 2	0.43105	

The following sentence has been added to the caption of Figure 5: "Absolute differences in peak intensities should be ascribed to the specific experimental conditions."

- The authors argue that Sr segregation is related to a total energy argument, which can be avoided/reduced by chemical doping. Does it hence take place independent of ambient oxygen pressure? Can the authors comment how this compares to segregation phenomena triggered by the (partial) Schottky equilibrium such as in the case of SrTiO₃, where Sr segregation is triggered primarily at high oxygen pressure in order to form Sr vacancy defects?

Unlike SrTiO₃, Sr is present in the LSM lattice as an aliovalent impurity, i.e. it determines lattice distortion and the presence of an electrostatic charge. These have been ascribed as the main driving forces of Sr segregation in LSM and related MIEC materials by extensive literature (c.f. especially works from Yildiz group: Lee et al., J. Am. Chem. Soc. 2013, 135, 7909-7925 and Kim et al., J. Am. Chem. Soc. 2020, 142, 7, 3548–3563). Sr segregation can be interpreted as resulting from a Schottky-type reaction in which elastic and electrostatic contributions determine a decrease in the defect formation enthalpy. In our work DFT modelling, we focused on the elastic contribution as it arguably sets the main difference between LSM and VAN. DFT calculations confirm significant overbonding for the Sr atom in LSM (i.e., reduced lattice stability), thus justifying our approach.

As far as the oxygen partial pressure dependence is concerned, one can expect a complex situation in which the concentration of oxygen vacancies in the bulk, on the surface as well as the reactivity of surface Sr with gas oxygen determine the final tendency for Sr to segregate. Interestingly, this has direct implications on the behavior of our VAN functional layer in full SOC devices, where the oxygen chemical

potential (or, the equivalent partial pressure) changes even locally and depending on the cell operational point. In order to address the behavior of our VANs in real devices, the authors are now performing a dedicated study (unpublished and therefore shown here confidentially) where VAN functional layers are employed in state-of-the-art cells provided by the most important SOFC manufacturer in Europe (Solid Power) showing excellent preliminary results. In **Figure R2**, we report the outcome of a long-term degradation test, carried out on an anode-supported full SOFC operating at 750 °C and with an applied current density of 0.3 A*cm⁻². The test reveals an outstanding stability for the cells, which after **750 hours** of operation still maintains voltage (power) values which are better than what measured at the beginning of the experiment. Such an important result suggests that VANs are effective in improving not just the electrode thermal stability (i.e. in air and without applied bias) but also its electrochemical behavior under cathodic polarization (or, under decreased equivalent oxygen partial pressure).

Figure R2. Long-term durability test for an anode-supported SOFC in which the cathode is formed by PLD barrier layer / 100 nm VAN functional layer / airbrushed LSCF electrode and current collector. The test is carried out using dry H₂ as a fuel.

- Finally, the authors bring in a ‘high-entropy’-argument at the end of their paper. Can the authors comment if in that case a solid solution would be preferred against the coherently ordered VANs in order to further increase entropic energy?

As pointed out by McCormack et al. in a recent review on HEOs (Acta Mater. 202, 1-21, 2021), when dealing with the energy terms which lead to lattice stabilization, interface energy plays a prominent role. We believe therefore that the presence of a high interface density together with the small volumes under consideration are important factors for the observed thermal stability of the VANs.

At page 19, we modified a sentence in order to clarify this point: “We assign such an increased solubility limit to a large contribution of surface energy to the final free energy, which is maximized here by the very high interface density, and to the small volumes under consideration;”

It is also important to point out that, after the submission of the present manuscript, Yang et al. (J. Power Sources, 482, 228959, 2021) have reported suppressed Sr segregation for highly substituted (high entropy) LSM. In this case however, no remarkable increase in the electrochemical properties with

respect to LSM was observed. These results show that, although entropy-driven stabilization can be achieved also for single phase LSM, our approach based on the coexistence between local disorder (in the LSM phase) and structural coherence (owing to the VAN structure) is key for the realization of a material with superior performance.

Such a reference is now commented in the revised version of the manuscript at page 20: “Entropy-induced lattice stabilization has been recently demonstrated for bulk LSM; In such a case however, no increase in the electrochemical properties has been found with respect to the LSM parent compound, highlighting the relevance of our VAN strategy for combining high thermal stability – owing to local high-entropy in the LSM phase – and strongly enhanced ORR kinetics due to the intimate phase alternation and off-plane geometry.”

The following reference has been added: Y. Yang, H. Bao, H. Ni, X. Ou, S. Wang, B. Lin, P. Feng and Y. Ling, A novel facile strategy to suppress Sr segregation for high-entropy stabilized $\text{La}_{0.8}\text{Sr}_{0.2}\text{MnO}_{3-\delta}$ cathode, *J. Power Sources*, 2021, 482, 228959.

Reviewer #1 (Remarks to the Author):

The authors have considered all of my questions and comments and have improved the manuscript accordingly. From my point of view the manuscript is now suitable for publication in Nature Communications.

Reviewer #2 (Remarks to the Author):

The manuscript by Baiutti et al. has been significantly improved upon revision and I recommend publication of the manuscript (in line with my previous recommendation).

I have to emphasize that I highly appreciate the excellent and detailed response letter together with the thorough revisions. The authors present clear arguments and have addressed my comment with great accuracy and care. I have enjoyed reading the author response as well as the revised manuscript. I will be happy to use and cite this work once it is available online.

I have only a few remaining comments, the authors should take into account before publication:

- Given the clarity of the authors' argumentation in the response letter, I would suggest to include all relevant aspects they mention in the manuscript (as long as the authors do not fear to lose focus) and to include also all the literature mentioned in their argumentation in the response letter, as they are obviously important for the paper.

- With respect to the 'high-entropy' discussion, I still feel that a little bit of additional explanation will be helpful for the reader. Particularly, since this attribute was now moved to the title – in my eyes, the authors argue based on an energy argument. I surely share this idea, that increasing surface energy contribution can indeed stabilize the lattice and potentially overcome solubility limits. But this is an energy argument, rather than an entropy argument, right?
Or is the idea to further stabilize 'a high entropy compound such as LSM' by additionally inserting interfaces? For the VAN itself I would always expect that in terms of entropy the solid solution is favored as the ordered VAN structure reflects only a subset of possible configurations of a solid solution? A few lines further clarifying the authors meaning would be helpful.

- Finally, thermodynamically, it is still a bit surprising for me that Sr shall be reincorporated into LSM during thermal treatment, as both elastic energies (as shown by the authors and MIT group) as well as Schottky-equilibria (proposed by the reviewer) should similarly favor Sr to stay outside LSM when exposed to oxygen. I can however accept the authors empirical statement together with the cited literature, given that this is also only a side argument in the paper.

Reviewer #2 (Remarks to the Author):

The manuscript by Baiutti et al. has been significantly improved upon revision and I recommend publication of the manuscript (in line with my previous recommendation).

I have to emphasize that I highly appreciate the excellent and detailed response letter together with the thorough revisions. The authors present clear arguments and have addressed my comment with great accuracy and care. I have enjoyed reading the author response as well as the revised manuscript. I will be happy to use and cite this work once it is available online.

Once again we would like to thank the reviewer for appreciating our work and we would like to acknowledge the outstanding quality of the review provided. We have modified the manuscript according to the reviewer's suggestion. The new changes are highlighted in green in the manuscript.

I have only a few remaining comments, the authors should take into account before publication:

- Given the clarity of the authors' argumentation in the response letter, I would suggest to include all relevant aspects they mention in the manuscript (as long as the authors do not fear to loose focus) and to include also all the literature mentioned in their argumentation in the response letter, as they are obviously important for the paper.

Following the reviewer's suggestion, we have included the following paragraphs to the main text (and related references).

On page 9:

"Please note that the experimental data do not suggest any singularity in the oxygen diffusivity at the interface, either because not present or because of a very limited effect on the behavior of the VAN thin film. For these reasons, the FEM model does not include any nanoscopic parameters for describing oxygen transport at the LSM/SDC interface, which is modeled as an abrupt change of diffusivity in the two materials (continuous layer)."

On page 12:

"Notably, these considerations are perfectly in line with previous works from the co-authors (Chiabrera et al., Adv. Mater. 2019, 31, 1805360 and Chiabrera et al., Solid State Ionics 299, 70-77, 2017),^{46,57} in which a decrease in ORR activation energy was measured for B-site deficient LSM. There, it was concluded that lowering the Mn concentration generates a reduction of oxygen vacancy formation energy and a consequent enhancement of the ORR kinetics. Interestingly, the final ORR activation energy value for strongly B-site deficient LSM is in good agreement with the value of our VANs (≈ 1.8 eV):⁵⁸ In both cases, introducing mobile oxygen vacancies in the system greatly enhances the ORR kinetics. In VANs, a better electrochemical performance is ensured by the higher mobility and concentration of oxygen vacancies in the SDC phase."

57 F. Chiabrera, A. Morata, M. Pacios and A. Tarancón, *Solid State Ionics*, 2017, **299**, 70–77.

58 F. M. Chiabrera, *University of Barcelona, PhD thesis 2019*.

On page 13:

“In our case, for SDC and LSM nominal acceptor concentration is 20% at.; From here, one can estimate the Debye length to be ≈ 1 nm,⁵⁹ i.e. at least one order of magnitude lower than the pillar width.

As far as the in-plane current percolation is concerned, although a perfect VAN structure should not, in principle, allow for electronic current transport, one can imagine that the (yet typically negligible) electronic conductivity of the SDC under working conditions may be sufficient to ensure a certain in-plane percolation owing to the very small pillar width (≈ 10 nm). Current collection for the electrochemical experiments was in any case ensured by the utilization of porous Au paste (see also Experimental section), which allows the electrode to be partially in contact with the gas atmosphere while offering a closely spaced percolating path thus limiting in-plane potential drops. Such an approach has been extensively used in the past for dense and porous thin film electrodes.^{62,63”}

60 F. Gunkel, R. Waser, A. H. H. Ramadan, R. A. De Souza, S. Hoffmann-Eifert and R. Dittmann, *Phys. Rev. B*, 2016, **93**, 245431.

61 S. Kim, S. K. Kim, S. Khodorov, J. Maier and I. Lubomirsky, *Phys. Chem. Chem. Phys.*, 2016, **18**, 3023–3031.

62 D. Chen, S. R. Bishop and H. L. Tuller, *J. Electroceramics*, 2012, **28**, 62–69.

63 M. P. Wells, A. J. Lovett, T. Chalklen, F. Baiutti, A. Tarancón, X. Wang, J. Ding, H. Wang, S. Kar-Narayan, M. Acosta and J. L. Macmanus-Driscoll, *ACS Appl. Mater. Interfaces*, 2021, **13**, 4117–4125.

- With respect to the ‘high-entropy’ discussion, I still feel that a little bit of additional explanation will be helpful for the reader. Particularly, since this attribute was now moved to the title – in my eyes, the authors argue based on an energy argument. I surely share this idea, that increasing surface energy contribution can indeed stabilize the lattice and potentially overcome solubility limits. But this is an energy argument, rather than an entropy argument, right?

Or is the idea to further stabilize ‘a high entropy compound such as LSM’ by additionally inserting interfaces? For the VAN itself I would always expect that in terms of entropy the solid solution is favored as the ordered VAN structure reflects only a subset of possible configurations of a solid solution? A few lines further clarifying the authors meaning would be helpful.

We agree on the reviewer that the discussion on the role of high entropy deserves further clarification. Indeed our point is based on energy and follows well-accepted recent literature (lead by the Yildiz group at MIT), assigning one of the driving forces to Sr segregation to elastic energy minimization. Our approach has been focused on analyzing how a homogeneous dispersion of Ce and Sm atom species in the perovskite A-site affects the local strain field around the Sr atom, or in other words how high configurational entropy is linked to the elastic driving force to surface segregation. Although we agree

that a more extensive analysis, including also the role of interfaces, could be interesting, we believe that the indications given by our DFT simulations are clear in attributing a prominent role in the observed stability to cationic intermixing via local lattice distortion.

We have carried out a thorough revision of the DFT section in the paper by moving/modifying several paragraphs and wording, as one can see in the corrected version of the manuscript. The main additions are the following:

“Here, we directly evaluate the effect of the homogeneously dispersed multiple atom species in the perovskite A-site on the local strain field around the Sr atom, i.e. on the elastic driving force for atom surface segregation.”

“In other words, the increase of configurational entropy dictates the enhancement of the LSM stability ~~the excellent long-term behavior of the VAN structure~~ by triggering a local lattice distortion. Besides, in agreement with recent theoretical works on high entropy oxides,⁷³ one may consider also a stabilizing effect of the interfaces, of i.e. ~~to a large~~ the contribution of surface energy to the final free energy, which is maximized here by the very high interface density, ~~and to the small volumes under consideration~~”

- Finally, thermodynamically, it is still a bit surprising for me that Sr shall be reincorporated into LSM during thermal treatment, as both elastic energies (as shown by the authors and MIT group) as well as Schottky-equilibria (proposed by the reviewer) should similarly favor Sr to stay outside LSM when exposed to oxygen. I can however accept the authors empirical statement together with the cited literature, given that this is also only a side argument in the paper.

We thank the reviewer for the interesting observation, which we will certainly take into account for further studies.